# Original COVID-19 priming regimen impacts the immunogenicity of bivalent BA.1 and BA.5 boosters

Luca M. Zaeck [1,14], Ngoc H. Tan [2,14], Wim J. R. Rietdijk [2], Daryl Geers[1], Roos S. G. Sablerolles[2], Susanne Bogers[1], Laura L. A. van Dijk [1], Lennert Gommers[1], Leanne P. M. van Leeuwen[1], Sharona Rugebregt[1], Abraham Goorhuis[3,4], Douwe F. Postma [5], Leo G. Visser[6], Virgil A. S. H. Dalm[7,8], Melvin Lafeber[9], Neeltje A. Kootstra [10], Anke L. W. Huckriede[11], Bart L. Haagmans [1], Debbie van Baarle[11,12], Marion P. G. Koopmans [1], SWITCH-ON Research Group*, P. Hugo M. van der Kuy [2,15], Corine H. GeurtsvanKessel [1,15] ✉ & Rory D. de Vries [1,15]

Waning antibody responses after COVID-19 vaccination combined with the emergence of the SARS-CoV-2 Omicron lineage led to reduced vaccine effectiveness. As a countermeasure, bivalent mRNA-based booster vaccines encoding the ancestral spike protein in combination with that of Omicron BA.1 or BA.5 were introduced. Since then, different BA.2-descendent lineages have become dominant, such as XBB.1.5, JN.1, or EG.5.1. Here, we report post-hoc analyses of data from the SWITCH-ON study, assessing how different COVID-19 priming regimens affect the immunogenicity of bivalent booster vaccinations and breakthrough infections (NCT05471440). BA.1 and BA.5 bivalent vaccines boosted neutralizing antibodies and T-cells up to 3 months after boost; however, cross-neutralization of XBB.1.5 was poor. Interestingly, different combinations of prime-boost regimens induced divergent responses: participants primed with Ad26.COV2.S developed lower binding antibody levels after bivalent boost while neutralization and T-cell responses were similar to mRNA-based primed participants. In contrast, the breadth of neutralization was higher in mRNA-primed and bivalent BA.5 boosted participants. Combined, our data further support the current use of monovalent vaccines based on circulating strains when vaccinating risk groups, as recently recommended by the WHO. We emphasize the importance of the continuous assessment of immune responses targeting circulating variants to guide future COVID-19 vaccination policies.

Vaccination against coronavirus disease-2019 (COVID-19) provides protection against infection, hospitalization, and mortality[1,2]. However, the ongoing waning of severe acute respiratory syndrome coronavirus-2 (SARS-CoV-2)-specific immune responses and the continuous evolution of antigenically distinct variants result in an overall reduction of vaccine effectiveness[3]. The Omicron BA.2-descendent variants such as XBB.1.5 and BA.2.86, that circulated at the time of this study, were the most immune evasive variants at that

A full list of affiliations appears at the end of the paper. *A list of authors and their affiliations appears at the end of the paper.
✉e-mail: c.geurtsvankessel@erasmusmc.nl

point[4–6]. This is an ongoing arms race: adapted vaccines are required to retain effective protection on a population level, especially in vulnerable at-risk patients, in the face of new emerging variants. To this end, mRNA-based bivalent vaccines incorporating an Omicron BA.1 or BA.5 spike (S) protein in combination with the ancestral S were introduced in 2022[7,8].

While the mRNA-based vaccines BNT162b2 and mRNA-1273 were initially shown to have higher vaccine efficacy over adenovirus-vectored vaccines (Ad26.COV2.S and ChAdOx1-S) in a primary vaccination series[3,9], it is not known whether different original priming regimens have a long-lasting imprinting effect on the magnitude, durability, or breadth of the SARS-CoV-2-specific immune response[10]. Heterologous COVID-19 vaccination with different vaccine platforms but the same S antigen was demonstrated to be at least non-inferior regarding immunogenicity when compared to homologous priming with either mRNA-based or adenovirus-based vaccines alone[11–13]. Shaping of the immune response as a consequence of exposure to different S antigens was mostly studied in the context of hybrid immunity, a combination of vaccination and infection. These studies showed evidence for serological imprinting to the ancestral S protein, but also the induction of variant-specific immune responses[14–16].

The SWITCH-ON trial[17,18] aimed to evaluate the mRNA-based bivalent BA.1 and BA.5 booster vaccines developed by BioNTech/Pfizer (BNT162b2 Omicron BA.1/BA.5) or Moderna (mRNA-1273.214 and mRNA-1273.222) against the background of different priming regimens (mRNA-based or Ad26.COV2.S), by addressing three crucial questions: (1) How immunogenic are Omicron BA.1 or BA.5 bivalent booster vaccines? (2) Do BA.1 or BA.5 bivalent booster vaccines differ in the induction of broad neutralizing antibody responses, including adequate neutralization of XBB-descendent variants? (3) How do immune responses among different original priming vaccination regimens evolve over time and what can we learn for the future?

## Results

### Study design and baseline characteristics

A total of 434 healthcare workers (HCW) were included in the SWITCH-ON trial after screening of 592 potential participants (Fig. 1, baseline characteristics in Supplementary Tables S1 and S2). HCW received either Ad26.COV2.S or an mRNA-based (mRNA-1273 or BNT162b2) priming vaccination regimen, followed by at least one mRNA-based booster vaccination before inclusion in this study. The SWITCH-ON trial comprised two groups to which the participants were randomly assigned: (1) a direct boost group (DB) (n = 219) or (2) a postponed boost (PPB) group (n = 183). Participants in the DB group were vaccinated in October 2022 with an Omicron BA.1 bivalent vaccine (BNT162b2 Omicron BA.1 or mRNA-1273.214); participants in the PPB group were vaccinated in December 2022 with an Omicron BA.5 bivalent vaccine (BNT162b2 Omicron BA.5 or mRNA-1273.222). Samples were collected before bivalent vaccination, at 7 and 28 days post-vaccination, and at approximately 3 months post-vaccination (Supplementary Fig. S1). As the performance of interferon-gamma (IFN-γ) release assays (IGRAs) and ELISpots was demonstrated to be comparable in healthy individuals[19,20], we chose to assess S-specific T-cell responses by IGRA to have a scalable, robust, and comparable platform across all university medical centers in our study. No formal statistical tests were performed to test for differences within or between groups

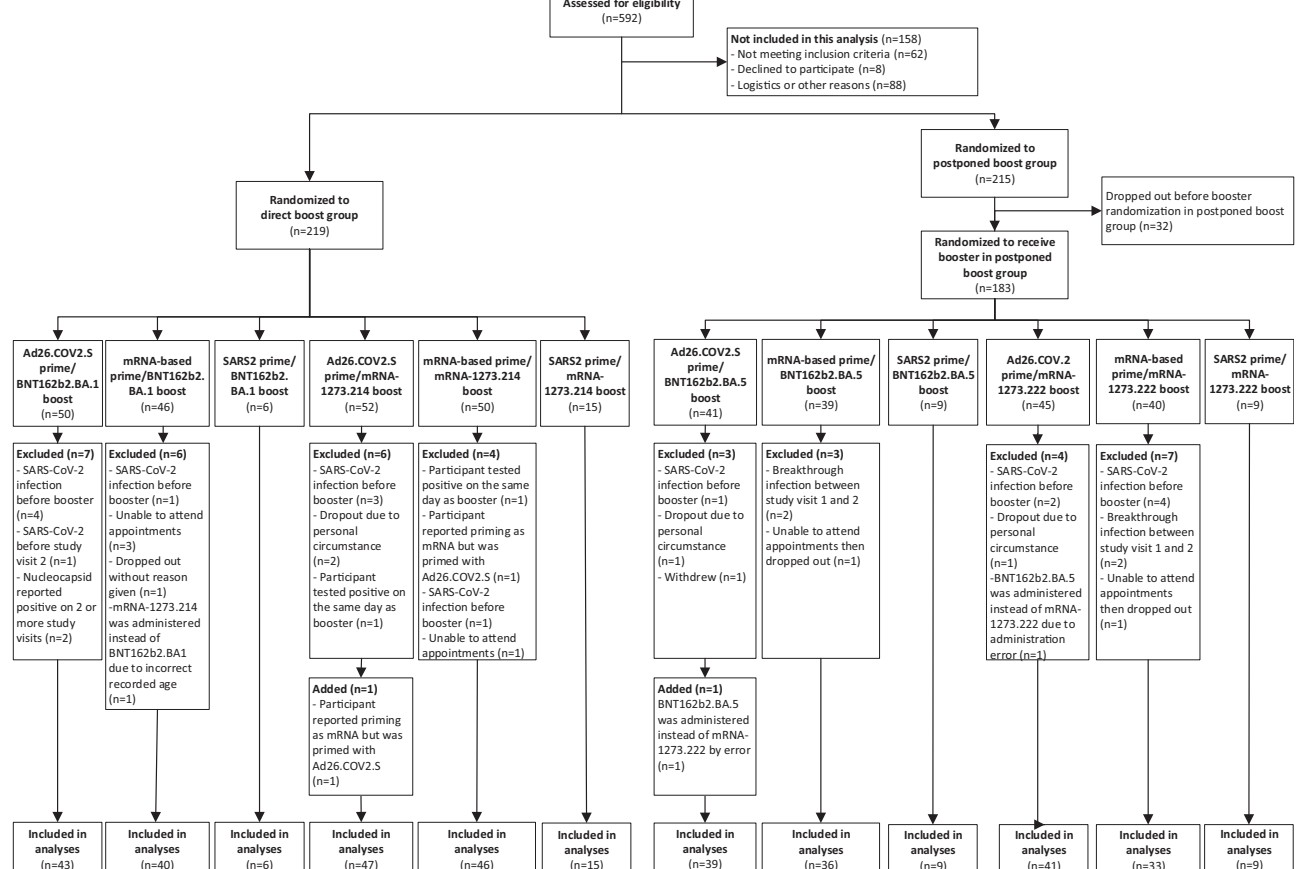

**Fig. 1 | SWITCH-ON trial enrollment.** A total of 592 healthcare workers (HCW) were screened for eligibility. Before inclusion in this study, HCW received either Ad26.COV2.S or an mRNA-based (mRNA-1273 or BNT162b2) priming vaccination regimen, followed by at least one mRNA-based booster vaccination. Of the 592 HCW, 434 were included and randomized 1:1 to the direct boost (n = 219) or the postponed boost (n = 215) group. Following dropouts, a total of 183 HCW received an Omicron BA.5 bivalent vaccine in the postponed boost group.

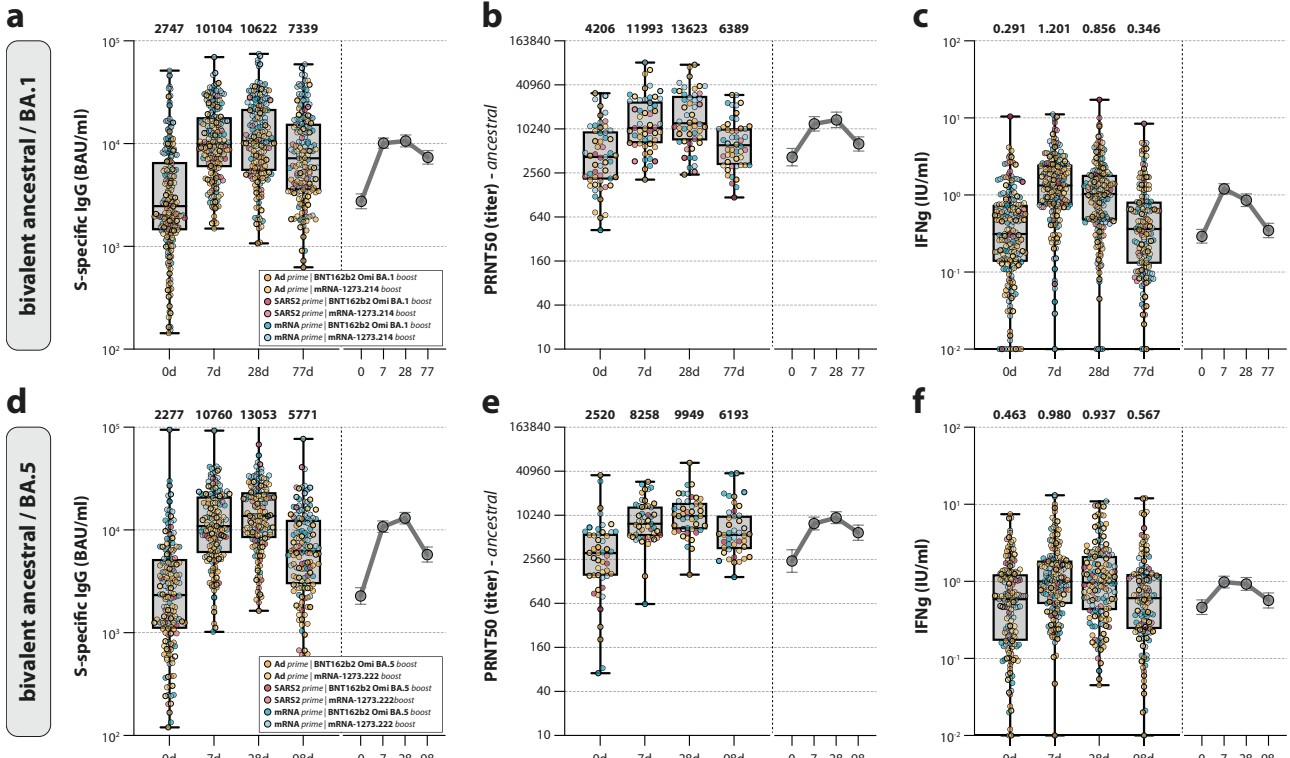

**Fig. 2 | Antibody and T-cell responses after bivalent booster vaccination.**
**a–f** Detection of (ancestral) spike (S)-specific binding IgG antibodies (**a**, **d**), ancestral SARS-CoV-2 neutralizing antibodies (**b**, **e**), and T-cell responses measured by interferon-gamma (IFN-γ) release assay (IGRA) (**c**, **f**) after Omicron BA.1 (**a–c**) or BA.5 (**d–f**) bivalent booster vaccination at baseline, and 7 days, 28 days, and 3 months post-boost. Colors indicate the specific prime-boost regimen (orange = Ad26.COV2.S prime, BNT162b2 Omicron BA.1 or BA.5 boost; yellow = Ad26.COV2.S prime, mRNA-1273.214 or mRNA-1273.222 boost; dark red = SARS-CoV-2 infection prime, BNT162b2 Omicron BA.1 or BA.5 boost; light red = SARS-CoV-2 infection prime, mRNA-1273.214 or mRNA-1273.222 boost; dark blue = mRNA-based prime, BNT162b2 Omicron BA.1 or BA.5 boost; light blue = mRNA-based prime, mRNA-1273.214 or mRNA-1273.222 boost). Data are shown in box-and-whisker plots, with the horizontal lines indicating the median, the bounds of the boxes indicating the interquartile range (IQR), and the whiskers indicating the range. Bold numbers above the plots represent the respective geometric mean (titer) per timepoint. The line graphs next to each panel depict a time course of the respective geometric mean values with 95% confidence intervals. The number of biologically independent samples (serum or whole blood) used per assay is shown in Supplementary Table S4.

as we deviated from the original protocol in terms of pre-specified outcomes and a lower-than-anticipated sample size[18].

## Bivalent COVID-19 vaccines induce antibody and T-cell responses

The immunogenicity of Omicron BA.1 bivalent vaccines up to 28 days post-vaccination in the SWITCH-ON trial was reported previously[17]. Both S-specific IgG binding and neutralizing antibodies targeting ancestral SARS-CoV-2 increased within the first 28 days, with most of the increase occurring between days 0 and 7 (Fig. 2a, b). S-specific T-cell responses increased rapidly in the first 7 days post-vaccination and subsequently waned (Fig. 2c). At 3 months post-vaccination, all of the measured immune parameters had decreased in comparison to the previous study visit. Whereas antibodies did not yet wane to baseline levels, T-cell responses returned close to the baseline. The magnitude and kinetics of antibody and T-cell responses induced by Omicron BA.5 bivalent booster vaccination were comparable to the Omicron BA.1 bivalent boost, again with most of the increase occurring within the first 7 days (Fig. 2d–f). Overall, a comparable boost of (neutralizing) antibody and T-cell responses against ancestral SARS-CoV-2 was observed after either Omicron BA.1 or BA.5 bivalent boost, independent of the timing of vaccine administration.

## mRNA-based priming leads to higher binding antibody levels after bivalent boost

The two groups (DB, Omicron BA.1 bivalent boost; PPB, Omicron BA.5 bivalent boost) could each be subdivided into four subgroups, based on different priming and bivalent booster regimens: (1) Ad26.COV2.S prime and mRNA-1273.214 or mRNA-1273.222 boost, (2) Ad26.COV2.S prime and BNT162b2 Omicron BA.1 or BA.5 boost, (3) mRNA (mRNA-1273 or BNT162b2)-based prime and mRNA-1273.214 or mRNA-1273.222 boost, and (4) mRNA-based prime and BNT162b2 BA.1 or BA.5 boost (Supplementary Fig. S1). Notably, Omicron BA.1- or BA.5-boosted participants who had previously received an mRNA-based priming vaccination regimen consistently had higher levels of S-specific binding antibodies than those who received an Ad26.COV2.S priming (Fig. 3a, b, compare dark and light blue to orange and yellow). This effect of the original priming was not observed for ancestral SARS-CoV-2 neutralizing antibodies or T-cell responses (Fig. 3c–f).

When subdividing the two groups, we excluded participants primed with a SARS-CoV-2 infection before their priming vaccination. These participants with an infection as priming were analyzed separately; we observed kinetics that closely resembled those who received Ad26.COV2.S priming (Supplementary Fig. S2). S-specific binding antibodies of SARS-CoV-2 infection-primed individuals in particular were consistently lower when compared to the mRNA-based priming, while T-cell responses were more comparable. Identification of those who experienced a SARS-CoV-2 infection before the priming vaccination occurred post-hoc and consequently only includes a small number of samples.

Of specific interest, bivalent booster vaccination with mRNA-1273.214 or mRNA-1273.222 resulted in a larger increase of binding and neutralizing antibodies than boosting with their BNT162b2 counterparts did, indicating that these vaccines are more immunogenic

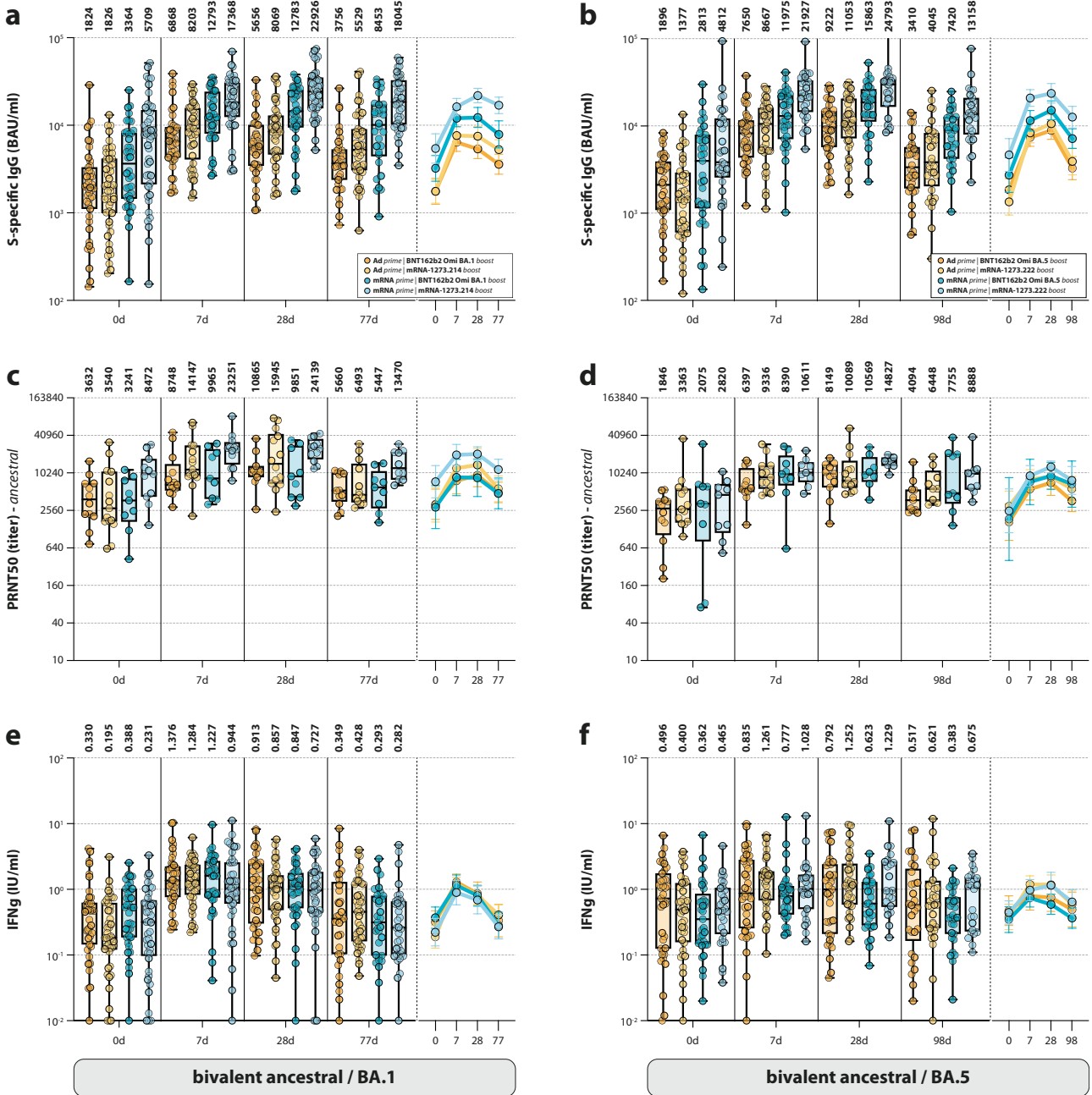

**Fig. 3 | Antibody and T-cell responses after different original priming and bivalent booster vaccinations. a–f** Detection of (ancestral) spike (S)-specific binding IgG antibodies (**a, b**), ancestral SARS-CoV-2 neutralizing antibodies (**c, d**), and T-cell responses measured by interferon-gamma (IFN-γ) release assay (IGRA) (**e, f**) based on the different combinations of original priming regimen after Omicron BA.1 (**a, c, e**) or BA.5 (**b, d, f**) bivalent booster vaccination at baseline, and 7 days, 28 days, and 3 months post-boost. Colors indicate the specific prime-boost regimen (orange = Ad26.COV2.S prime, BNT162b2 Omicron BA.1 or BA.5 boost; yellow = Ad26.COV2.S prime, mRNA-1273.214 or mRNA-1273.222 boost; dark

blue = mRNA-based prime, BNT162b2 Omicron BA.1 or BA.5 boost; light blue = mRNA-based prime, mRNA-1273.214 or mRNA-1273.222 boost). Data are shown in box-and-whisker plots, with the horizontal lines indicating the median, the bounds of the boxes indicating the interquartile range (IQR), and the whiskers indicating the range. Bold numbers above the plots represent the respective geometric mean (titer) per timepoint. The line graphs next to each panel depict a time course of the respective geometric mean values with 95% confidence intervals. The number of biologically independent samples (serum or whole blood) used per assay is shown in Supplementary Table S4.

(Fig. 3). For binding antibodies on 28 days post-vaccination, mRNA-1273.214-boosted participants had a 5.2-fold increase compared to a 4-fold increase in BNT162b2 Omicron BA.1-boosted participants. For the bivalent BA.5 counterparts, fold changes were 10.6-fold and 8.5 fold when comparing mRNA-1273.222-boosted participants with BNT162b2 Omicron BA.5-boosted participants (Fig. 4). These findings indicate that different prime-boost regimens lead to divergent immune responses.

## mRNA-based prime followed by BA.5 bivalent boost leads to broad neutralization

Neutralizing antibodies against relevant Omicron variants BA.1 and BA.5 (encoded by the vaccines), and XBB.1.5 (circulating at the time of the study) were measured to assess the breadth of the neutralization response (Fig. 5a, b). Comparable to ancestral SARS-CoV-2 neutralization, Omicron BA.1 and BA.5 neutralization was boosted by both the BA.1 and BA.5 bivalent booster vaccines; however, levels remained

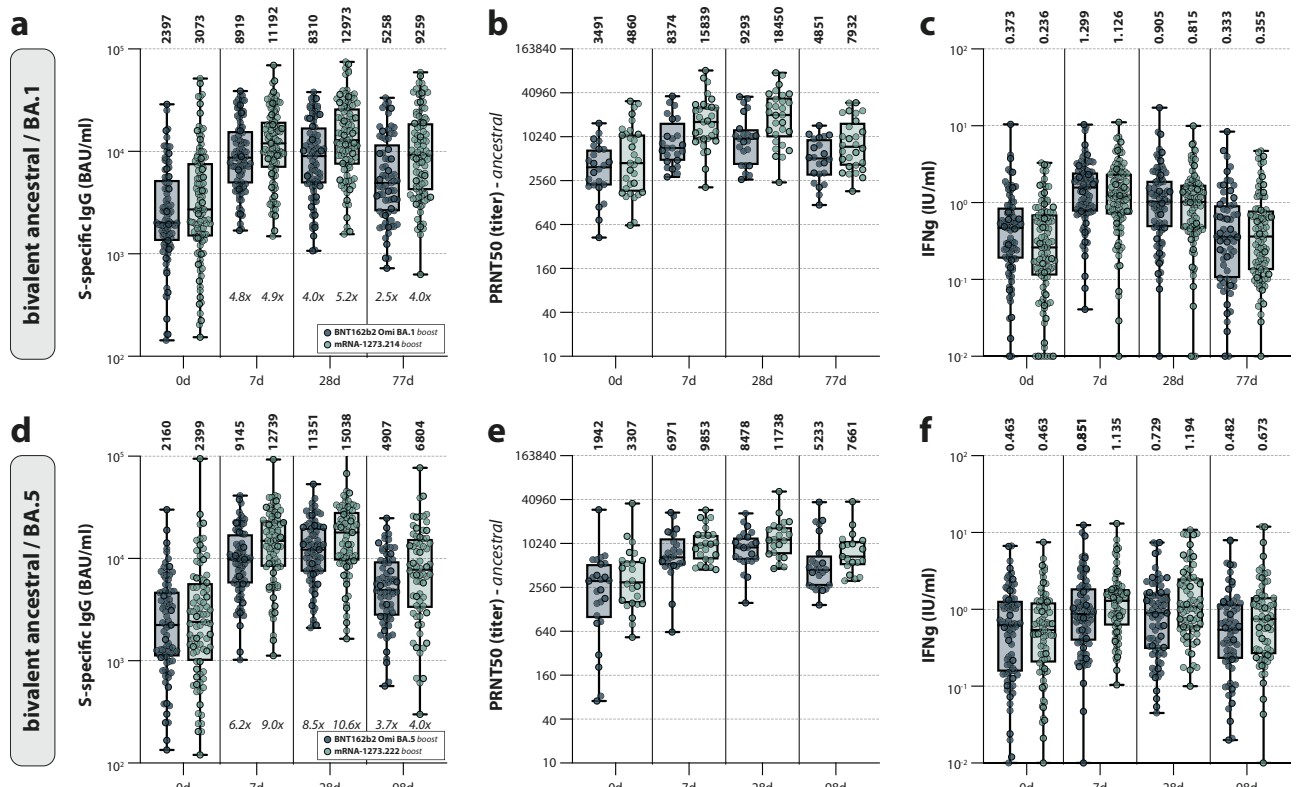

**Fig. 4 | Antibody and T-cell responses after Omicron BA.1/BA.5 bivalent booster vaccination separated by booster manufacturer. a–f** Detection of (ancestral) spike (S)-specific binding IgG antibodies (**a, d**), ancestral SARS-CoV-2 neutralizing antibodies (**b, e**), and T-cell responses measured by interferon-gamma (IFN-γ) release assay (IGRA) (**c, f**) after Omicron BA.1 (**a–c**) or BA.5 (**d–f**) bivalent booster vaccination with either BNT162b2 Omicron BA.1 or BA.5 (blue) or mRNA-1273.214 or mRNA-1273.222 (green) at baseline, and 7 days, 28 days, and 3 months post-boost.

Data are shown in box-and-whisker plots, with the horizontal lines indicating the median, the bounds of the boxes indicating the interquartile range (IQR), and the whiskers indicating the range. Bold numbers above the plots represent the respective geometric mean (titer) per timepoint. Italic numbers below the plots indicate fold changes relative to the baseline. The number of biologically independent samples (serum or whole blood) used per assay is shown in Supplementary Table S4.

below those for ancestral SARS-CoV-2 neutralization at all timepoints. At 3 months post-vaccination, waning of neutralizing antibodies was observed. Remarkably, when correlating ancestral- and variant-specific neutralizing antibody titers (Supplementary Fig. S3), it was clear that the waning of Omicron BA.1 and BA.5 neutralizing antibodies occurred at a slower rate compared to ancestral SARS-CoV-2 neutralizing antibodies. This was true for both individuals boosted with the bivalent Omicron BA.1 (Fig. 5c, e) or BA.5 vaccine (Fig. 5d, f). The circulating Omicron XBB.1.5 was poorly cross-neutralized at 3 months after a bivalent boost, irrespective of the different prime-boost regimens (Fig. 5a, b). In participants boosted with the bivalent Omicron BA.5 vaccine, a preferential boost of Omicron BA.5 neutralization was observed. This was not the case for Omicron BA.1 neutralizing antibodies in participants boosted with the bivalent Omicron BA.1 vaccine (Fig. 5g, compare orange with purple radar plot). When subdividing participants boosted with bivalent Omicron BA.5 in their respective prime-boost regimens, preferential boosting of Omicron BA.5 neutralization was predominantly visible in participants primed with an mRNA-based vaccine (Fig. 5h). Participants primed with Ad26.COV2.S retained a relatively narrow neutralizing response, despite receiving the bivalent Omicron BA.5 booster. These differences were not observed when measuring binding antibodies against different S variants. Binding levels to XBB.1.5 S protein were similar to binding levels to the BA.1 and BA.5 S protein (Supplementary Fig. S4a, b), and the binding antibody levels correlated well with neutralizing antibody levels (Supplementary Fig. S4c, d). Preferential boosting of BA.5-reactive binding antibodies after Omicron BA.5 bivalent vaccination was still observed (Supplementary Fig. 4e), but the increased breadth for

mRNA-primed individuals was not observed (Supplementary Fig. S4f, compare to Fig. 5h).

## Breakthrough infections lead to boosting of immune responses

In the PPB group, which was included in September 2022 but scheduled to receive the bivalent Omicron BA.5 vaccine in December 2022, 12 test-confirmed infections were detected before administration of the booster dose (Fig. 6a). While the respective variant the participants were infected with was not determined, the circulating variants at the time in the Netherlands were Omicron BA.5 and BQ.1[21]. These participants were subsequently excluded from the vaccination trajectory and analyzed separately as part of a natural infection-related sub-study. Breakthrough infection before bivalent vaccination boosted S-specific binding antibodies and T-cell responses. Binding antibody levels 7 days (GMT 3203 BAU/mL [95% CI 1983–5176]) and 28 days (GMT 4291 BAU/mL [95% CI 3242–5678]) post-infection (Fig. 6b) were lower than compared to the same time interval post-vaccination (7d: GMT 10,760 BAU/mL [95% CI 9463–12,235]; 28d: GMT 13,053 BAU/mL [95% CI 11,481–14,841], shown in Fig. 2d). However, T-cell responses and Omicron neutralizing antibodies were comparable to post-vaccination responses, although T-cell responses returned to baseline faster compared to post-vaccination (Fig. 6c, d). In addition, 57 breakthrough infections after administration of either bivalent Omicron BA.1 or BA.5 booster vaccination were detected through various methods (test-confirmed or detection of nucleocapsid-specific antibodies). Of these participants, samples collected prior to infection were included in the immunogenicity analyses. Notably, breakthrough infection after

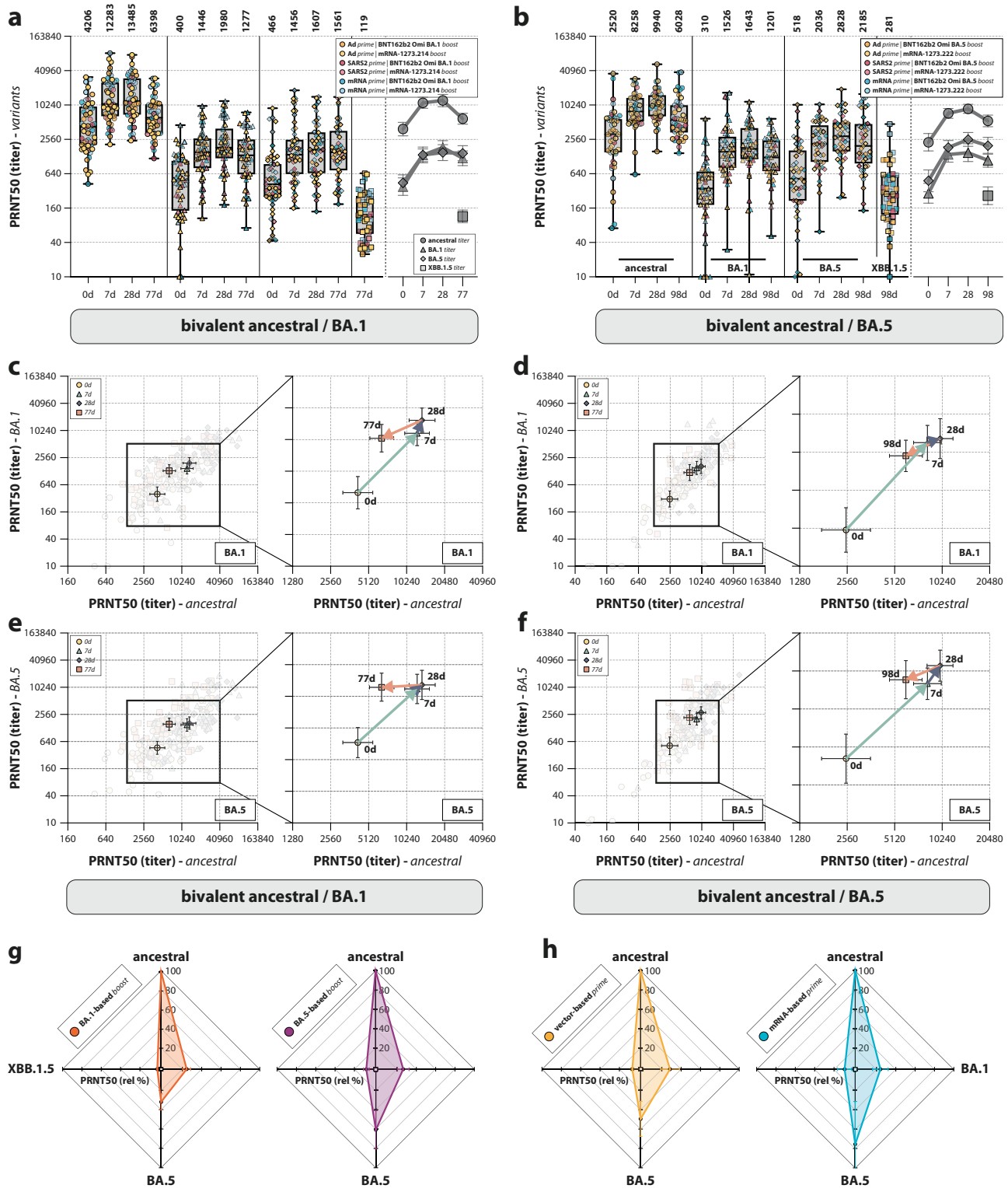

Omicron BA.1 or BA.5 bivalent boost did not result in an additional increase of antibody or T-cell responses in comparison to the already vaccine-induced levels (Supplementary Fig. S5).

## Discussion

Here, we report that Omicron BA.1 or BA.5 bivalent booster vaccination results in a rapid recall of humoral and cellular immune responses, which wane at 3 months post-vaccination. By simultaneously assessing multiple immune parameters, we found divergent immune responses after distinct COVID-19 vaccination regimens.

To our knowledge, this is the first study to assess the immunogenicity of bivalent vaccines in the context of different priming regimens. While the immunogenicity and boosting of SARS-CoV-2-specific immune responses by Omicron BA.1 or BA.5 bivalent vaccination was in line with previous studies[7,17,22], we find two important differences between bivalent-boosted participants primed with either Ad26.COV2.S or an mRNA-based vaccine: (1) mRNA-based priming leads to higher antibody levels upon boost, and (2) only a BA.5 bivalent boost led to broad neutralization profiles in mRNA-primed participants. This could be related to biological differences between the

**Fig. 5 | Breadth of the neutralizing antibody response after bivalent booster vaccination. a**, **b** Detection of neutralizing antibodies targeting ancestral SARS-CoV-2 and Omicron BA.1, BA.5, and XBB.1.5 variants after Omicron BA.1 (**a**) or BA.5 (**b**) bivalent booster vaccination at baseline, and 7 days, 28 days, and 3 months post-boost. Colors indicate the specific prime-boost regimen (orange = Ad26.COV2.S prime, BNT162b2 Omicron BA.1 or BA.5 boost; yellow = Ad26.COV2.S prime, mRNA-1273.214 or mRNA-1273.222 boost; dark red = SARS-CoV-2 infection prime, BNT162b2 Omicron BA.1 or BA.5 boost; light red = SARS-CoV-2 infection prime, mRNA-1273.214 or mRNA-1273.222 boost; dark blue = mRNA-based prime, BNT162b2 Omicron BA.1 or BA.5 boost; light blue = mRNA-based prime, mRNA-1273.214 or mRNA-1273.222 boost). **c**–**f** Correlation between 50% plaque reduction neutralization (PRNT$_{50}$) titers against ancestral SARS-CoV-2 and the Omicron BA.1 (**c**, **d**) or BA.5 (**e**, **f**) variants over time after Omicron BA.1 (**c**, **e**) or BA.5 (**d**, **f**) vaccination at baseline, and 7 days, 28 days, and 3 months post-boost. Colored symbols indicate the specific timepoints (yellow = baseline [0 d]; teal = 7 d; purple = 28 d;

orange = 77 d [c,e]/98 d [**d**, **f**]). The arrows connect the correlated geometric means (+95% confidence intervals [CI]) per timepoint and visualize the neutralization kinetics. **g**, **h** Radar plots depicting the variant-specific PRNT$_{50}$ titers relative to ancestral SARS-CoV-2 neutralization (set to 100%) after vaccination with bivalent Omicron BA.1 or BA.5. The plots are grouped either by the administered Omicron BA.1 (orange) or BA.5 (purple) bivalent booster vaccination (**g**) or the original priming regimen (vector-based = yellow; mRNA-based = blue) after Omicron BA.5 bivalent vaccination (**h**). Data in (**a**, **b**) are shown in box-and-whisker plots, with the horizontal lines indicating the median, the bounds of the boxes indicating the interquartile range (IQR), and the whiskers indicating the range. Bold numbers above the plots represent the respective geometric mean (titer) per timepoint. The line graphs next to each panel depict a time course of the respective geometric mean values with 95% confidence intervals. The number of biologically independent sera is shown in Supplementary Table S4.

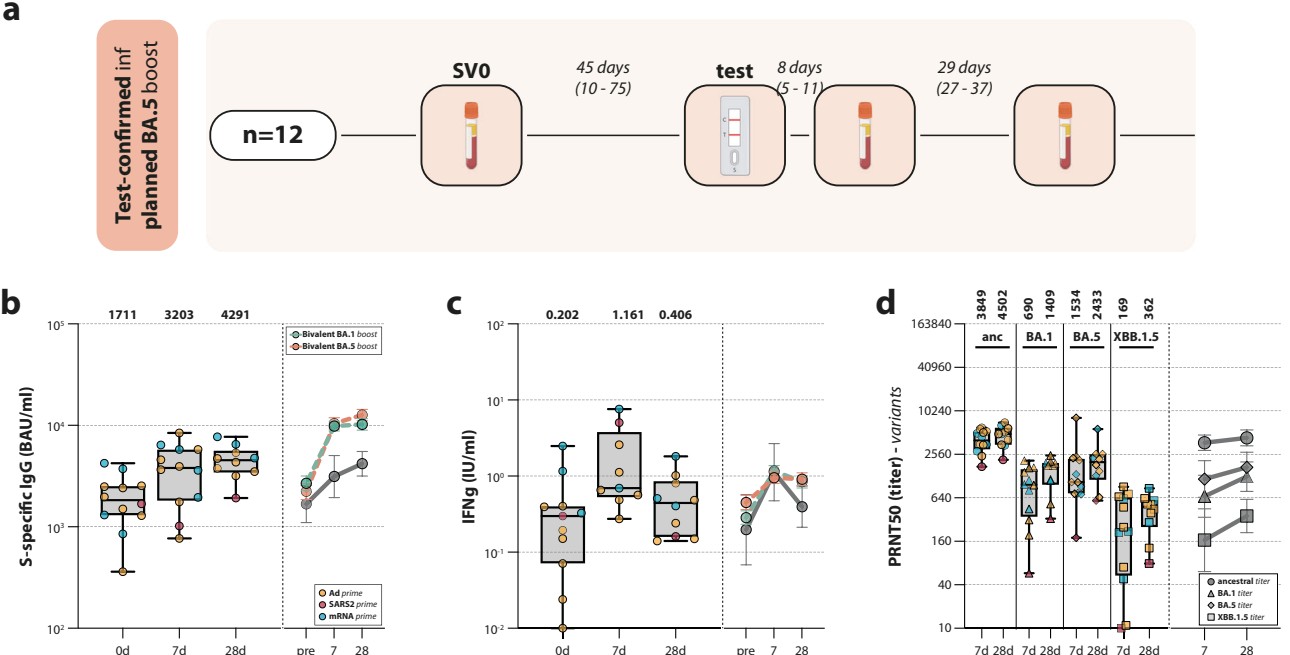

**Fig. 6 | Antibody and T-cell responses after breakthrough infection. a** Sampling procedure for participants in the postponed boost group who had a breakthrough infection before their intended vaccination with the bivalent Omicron BA.5 booster vaccine (*n* = 12). They were subsequently excluded from the vaccination trajectory and invited to participate in a sub-study on the immunogenicity of natural SARS-CoV-2 infection. Samples were collected at enrollment, and 7 and 28 days after the participants tested positive. Created with BioRender.com. **b**–**d** Detection of (ancestral) S-specific binding IgG antibodies (**b**), T-cell responses measured by interferon-gamma (IFN-γ) release assay (IGRA) (**c**), and neutralizing antibodies targeting ancestral SARS-CoV-2 and Omicron BA.1, BA.5, and XBB.1.5 variants (**d**) before, and 7 and 28 days after breakthrough infection, which was contracted

before intended vaccination with the bivalent Omicron BA.5 booster vaccine (yellow = Ad26.COV2.S prime; red = SARS-CoV-2 infection prime; blue = mRNA-based prime). Data are shown in box-and-whisker plots, with the horizontal lines indicating the median, the bounds of the boxes indicating the interquartile ranges (IQR), and the whiskers indicating the range. Bold numbers above the plots represent the respective geometric mean (titer) per timepoint. The line graphs next to each panel depict a time course of the respective geometric mean values with 95% confidence intervals. While the solid lines show the geometric mean values of the data from the box-and-whisker plots in the same panel, the dashed lines show reference values from comparable timepoints after either Omicron BA.1 (green) or BA.5 (orange) bivalent vaccination.

vaccine platforms, as it was already shown that the vaccine effectiveness for adenovirus-vectored vaccines was lower compared to mRNA-based vaccines[3,9]. In this context, it is important to emphasize that, given a number of vaccinations with possibly different vaccines and a varying number of exposures to SARS-CoV-2, the number and type of previous antigen exposures (i.e., exposure history) in a cohort can be a complex factor to account for at this point in the pandemic. However, as part of the SWITCH-ON study, we are in the unique position to have access to and account for the complete uninterrupted history of all SARS-CoV-2 antigen exposures for all individuals in our study (Supplementary Table S3). A post-hoc analysis of our cohort to identify individuals who were primed by a SARS-CoV-2 infection as their first

antigen exposure instead of vaccination indicated a trend towards lower antibody levels among infection-primed individuals when compared to mRNA-based priming.

When zooming in on the booster vaccines, mRNA-1273.214 and mRNA-1273.222 appeared more immunogenic than their BNT162b2 Omicron BA.1 and BA.5 counterparts. This supports a Moderna-funded retrospective cohort study, which reported greater effectiveness of mRNA-1273.222 compared with BNT162b2 Omicron BA.5 in preventing COVID-19-related hospitalizations and outpatient visits[23]. The differences in immunogenicity and efficacy between the BioNTech/Pfizer and Moderna vaccines are likely explained by differences in dose and/or antigen design. At 3 months post-bivalent booster vaccination, we

uniformly observed waning of all measured immune parameters, consistent with previous reports[24,25]. Interestingly, Omicron BA.1 and BA.5 neutralizing antibodies waned slower compared to ancestral SARS-CoV-2 neutralizing antibodies after bivalent boost. The number of antigen exposures could be underlying this observation; repeated exposure is thought to boost antibodies of the IgG4 subclass, potentially affecting functionality[26].

Neutralizing antibodies are assumed to be the immunological correlate of protection against symptomatic SARS-CoV-2 infection[9] and severe disease[27]. Based on this assumption, variant-modified booster vaccinations were predicted to offer an elevated level of protection[28]. While the overall effectiveness of Omicron bivalent vaccination has been described[23,29–31], we show that the cross-neutralization of the BA.2-descendent Omicron variant XBB.1.5, which was circulating at the time of the study, was poor after administration of either the Omicron BA.1 or the BA.5 bivalent booster vaccine, in line with previous reports[4–6]. Interestingly, this relative reduction of XBB.1.5 neutralizing antibodies compared to the level of neutralizing antibodies against ancestral SARS-CoV-2 or Omicron BA.1/BA.5 was not reflected by the binding antibody levels towards the XBB.1.5 S protein, which was comparable to BA.1/BA.5-binding antibody levels. This shows that the epitope changes of the XBB.1.5 S protein disproportionally affect antibody binding of its receptor-binding domain (RBD). It was recently demonstrated via RBD depletion experiments that the immune response following an Omicron BA.5 bivalent booster vaccination is primarily ancestral-specific and only cross-reactive towards BA.5, and that the concentrations of BA.5-specific antibodies are low[32]. This is in line with a report that spike-binding monoclonal antibodies derived memory B-cells isolated from individuals boosted with variant-modified mRNA vaccines (Beta/Delta bivalent or Omicron BA.1 monovalent) predominantly recognized the ancestral SARS-CoV-2 spike protein, with only a low frequency of de novo B-cells targeting variant-specific epitopes[33]. Similarly, induction of new antibody responses from naïve B cells was shown to be suppressed after sequential homologous boosting[15]. As demonstrated by the low XBB.1.5 neutralizing antibody levels at 3 months post-bivalent booster independent of the prime-boost regimen, it is logical to assume that these antibodies are even less cross-reactive with potential future lineages that are antigenically even more distinct[34,35]. Their reliance on the de novo induction of antigen-specific B cells to maintain vaccine effectiveness may be even larger. Consequently, this argues in favor of employing monovalent vaccines based on emerging lineages in subsequent vaccination campaigns, as recently recommended by the WHO[36].

The immunogenicity of SARS-CoV-2 breakthrough infections in comparison to booster vaccinations has not been extensively studied. Although we had a relatively small study size, and the variation between participants who had a breakthrough infection was large, we did find that breakthrough infections in participants who had been enrolled but not yet vaccinated led to a comparatively low boost of binding antibodies and a more rapid waning of T-cell responses. Furthermore, in participants who were vaccinated between 28 days and 3 months prior, no additional boost in S-specific responses was detected upon breakthrough, likely because antibody and T-cell responses were already relatively high. However, we only measured S-specific responses; breakthroughs could have potentially boosted immune responses to other antigens. Additionally, it is unknown how breakthrough infections with a certain variant affect protection from future infections with potentially different variants.

Combined, our data emphasize important lessons learned from the COVID-19 pandemic and associated vaccination strategies: (1) the original priming vaccination has an imprinting effect on the immune system that can still be observed after at least two mRNA-based booster vaccines, and (2) not all mRNA-based booster vaccines are equally immunogenic; in the SWITCH-ON trial only bivalent Omicron

BA.5 vaccination broadened the neutralizing antibody response, whereas the bivalent BA.1 vaccine did not. It is important to emphasize that this study was designed to investigate the magnitude, durability, and breadth of immune responses after an additional (bivalent) booster vaccination in a well-characterized cohort of healthcare workers. Consequently, our study cannot make any assertions on vaccine efficacy or other clinical outcomes of the imprinting-based altered immunogenicity of bivalent booster vaccinations. Our data support the recent vaccination advice from the WHO (as of May 2023)[36] to vaccinate risk groups with monovalent vaccines based on the circulating XBB.1-descendent lineage, as the current (bivalent) vaccines only induce limited cross-neutralization. Our data furthermore emphasize the importance of continuously evaluating immune responses and cross-reactivity with circulating variants to guide future COVID-19 vaccination policymaking.

## Methods

### Study design and participants

The SWITCH-ON study[17,18] is an ongoing multicenter, open-label, randomized controlled trial, which was conducted in accordance with the Declaration of Helsinki. Participants were randomized to either the direct boost group (DB) or the postponed boost group (PPB), who received a booster vaccination with an Omicron BA.1 or BA.5 bivalent vaccine in October or December 2022, respectively. This article reports the data for both study groups covering the period from the day of booster vaccination until 3 months post-vaccination. The study protocol (MEC-2022-0462) was approved by the Medical Ethics Committee of Erasmus University Medical Center (Rotterdam, the Netherlands), the sponsor site, and the local review boards of the other participating centers at the Amsterdam University Medical Centers, the Leiden University Medical Center, and the University Medical Center Groningen. Written informed consent was obtained from all study participants prior to the first study visit. There was no incentive or compensation for participation in the study. The study is registered with ClinicalTrials.gov (NCT05471440).

HCW between the age of 18–65 years were invited to join the SWITCH-ON trial from four academic hospitals in the Netherlands (Amsterdam University Medical Center, Erasmus Medical Center, Leiden University Medical Center, and University Medical Center Groningen). Eligible participants were primed with either one dose of an adenovirus-based (Ad26.COV2.S) or two doses of mRNA-based vaccine (BNT162b2 or mRNA-1273), and have received at least one mRNA-based booster vaccination. Prior SARS-CoV-2 infections were allowed; however, the last booster vaccination or SARS-CoV-2 infection had to have occurred at least 12 weeks before the bivalent booster was due, as per the advised interval between boosts from the National Institute for Public Health and the Environment (RIVM)[37]. Infection history was collected through a self-reported questionnaire. The full list of inclusion and exclusion criteria can be found in the study protocol[18]. A baseline table on the number of antigen exposures per original priming regimen and bivalent booster vaccination is provided in Supplementary Table S3. Sex or gender were not considered in the study design. Sex was collected in the study design, and reported in the baseline characteristics table. Our cohort comprises 73.6% female individuals, reflecting the female-dominant sex distribution among healthcare workers in the Netherlands[38]. A comparison of immunogenicity data from female and male participants is provided in Supplementary Fig. S6.

### Randomization and masking

All eligible participants were randomized using Castor software (v2024.1.0.3) to the DB (Omicron BA.1 bivalent boost) or PPB (Omicron BA.5 bivalent boost) group in a 1:1 ratio by block randomization with block sizes of 16 and 24. Due to the set-up of the study, it was not possible to blind participants from randomization. Therefore,

participants were informed about their group allocation prior to the first study visit. Randomization was completed by research assistants who were not involved in statistical analyses. Where necessary, sample selection was performed unblinded to allow equal sample numbers per subgroup. During data collection and analysis, researchers were blinded to sample information and were only exposed to study IDs.

## Procedures

Participants in the DB group received an Omicron BA.1 bivalent booster in October 2022. If participants were younger than 45 years old, BNT162b2 Omicron BA.1 was administered; mRNA-1273.214 was administered to participants 45 years and older. This age division was introduced as per advice from the RIVM because of safety concerns at the time regarding an increased myocarditis risk in young adults following administration of mRNA-1273.214[37]. Following consultation with the RIVM and the availability of additional published safety evidence, the age division was removed for the PPB group, and participants were randomized to receive the Omicron bivalent booster vaccination with either BNT162b2 Omicron BA.5 or mRNA-1273.222. Consequently, we deviated from the original study protocol by employing two vaccines per group (DB: BNT162b2 Omicron BA.1 (<45 years) and mRNA-1273.214 (≥45 years old); PPB [no separation by age]: BNT162b2 Omicron BA.5 and mRNA-1273.222) instead of one. As this affected the initial power calculations, sample sizes in the DB and PPB groups were adjusted to fit the new groups. In the DB and PPB groups, blood was taken during the first study visit (study visit 1, day 0). Additional blood samples were collected in subsequent study visits: study visit 2 (day $7 \pm 1$ days after boost), study visit 3 (day $28 \pm 2$ days after boost), and study visit 4 (day $90 \pm 14$ days after boost).

A baseline characteristics questionnaire was obtained after randomization to collect information about year of birth, biological sex, height, weight, ancestry, occupation, history of SARS-CoV-2 infection, and history of COVID-19 vaccination. A few days prior to each study visit, participants received a questionnaire to detect SARS-CoV-2 infections between the last and upcoming study visit. Via this infection questionnaire, we could identify participants who had an infection during the course of the study.

1.  If the infection occurred between the informed consent session and the first vaccination study visit, participants were invited to join a sub-study to analyze immunological response after natural infection and they would be excluded from vaccination trajectory. In this sub-study, blood samples would be collected at 7 and 28 days after participants tested positive (by at-home antigen test) for COVID-19 (Fig. 6).
2.  If the infection occurred between baseline and day 28 post-vaccination, no additional blood samples were taken as the mixed effect of natural infection and vaccination would be difficult to distinguish. These participants were excluded from all analyses.
3.  If the infection occurred between study visits on days 28 and 3 months post-vaccination, participants would be invited for additional blood sampling on days 7 and 28 after they had tested positive and remained in the study. Samples collected prior to infection were included in the immunogenicity analysis (Supplementary Fig. 5).

## Outcomes

According to the study protocol, the primary outcome was the fold change (i.e., the geometric mean ratio [GMR]) in antibody response between baseline and 28 days after boost in the DB group. Secondary outcomes were fast response, S-specific T-cell response, and levels of neutralizing antibodies[17,18]. Primary and secondary outcomes (immunogenicity of Omicron BA.1 bivalent vaccines up to 28 days post-vaccination) were reported previously[17]. Here, we report post-hoc analyses

based on data from the SWITCH-ON study. As we show observational data on the magnitude and quality of the immunological response, a descriptive approach was used to describe the immunogenicity of bivalent booster vaccinations over the period of 3 months following vaccination. We measured S-specific IgG binding antibodies, S-specific T-cell responses, and neutralization of the ancestral, BA.1, BA.5, and XBB.1.5 variants. Similar parameters were analyzed in the infection sub-study.

## Identification of recent SARS-CoV-2 infection

Infections were either identified via self-reporting of participants following a positive test result in an at-home antigen test or the detection of SARS-CoV-2 nucleocapsid (N)-specific antibodies. N-specific antibodies were measured at baseline and at 3 months post-boost using the Abbott SARS-CoV-2 IgG assay (Abbott, #06R-86-22) following the manufacturer's instructions. N-specific antibody levels were expressed in a signal-to-cut-off (S/CO) ratio and the manufacturer-recommended cut-off for positivity of ≥1.4 S/CO was used. If participants had detectable N-specific antibodies at 3 months post-boost, the other timepoints at 7 and 28 days post-boost were also tested to narrow down the moment of infection. All samples from the timepoint N-specific antibodies were detectable (or increased at least two-fold) and onwards were excluded from the immunogenicity analyses of bivalent booster vaccinations.

## Detection of SARS-CoV-2 S1-specific IgG antibodies

S1-specific antibodies were measured as previously described[39] by Liaison SARS-CoV-2 TrimericS IgG assay (DiaSorin, #311510). The lower limit of detection (LLoD) was 4.81 BAU/mL and the cut-off for positivity was 33.8 BAU/mL, according to the manufacturer's instructions.

## Detection of SARS-CoV-2 neutralizing antibodies

Serum samples were tested for the presence of neutralizing antibodies against ancestral SARS-CoV-2, and the Omicron BA.1, BA.5, and XBB.1.5 variants in a plaque reduction neutralization test (PRNT) as previously described[17]. Viruses were cultured from clinical material and sequences were confirmed by next-generation sequencing: D614G (ancestral; GISAID: hCov-19/Netherlands/ZH-EMC-2498), Omicron BA.1 (GISAID: hCoV-19/Netherlands/LI-SQD-01032/2022), Omicron BA.5 (EVAg: 010V-04723; hCovN19/Netherlands/ZHNEMCN5892), and Omicron XBB.1.5 (GISAID: hCov-19/Netherlands/NH-EMC-5667). The human airway Calu-3 cell line (ATCC HTB-55) was used to grow virus stocks and to conduct PRNT. Calu-3 cells were cultured in OptiMEM supplemented with GlutaMAX (Gibco, #51985-026), penicillin and streptomycin (100 units/mL and 0.1 mg/mL, respectively, Capricorn Scientific, #PS-B), and 10% fetal bovine serum (FBS; Sigma, #F7524). Briefly, heat-inactivated sera were two-fold serially diluted in OptiMEM without FBS. The dilutions range were based on the respective variant and the S-specific binding antibody level: ancestral SARS-CoV-2 (<1500 BAU/mL: 1:10–1:1280; 1500–6000 BAU/mL: 1:80–1:10,240; >6000 BAU/mL: 1:640 – 1:81,920), Omicron BA.1/BA.5 variants (<6000 BAU/mL: 1:10–1:1280; >6000 BAU/mL: 1:80–1: 10,240), Omicron XBB.1.5 variant (<6000 BAU/mL: 1:10–1:1280; >6000 BAU/mL: 1:40–1: 5120). Four hundred PFU of either SARS-CoV-2 variant in an equal volume of OptiMEM medium were added to the diluted sera and incubated at 37 °C for 1 h. The antibody-virus mix was then transferred to Calu-3 cells and incubated at 37 °C for 8 h. Afterwards, the cells were fixed in 10% neutral-buffered formalin, permeabilized in 70% ethanol, and the plaques stained with a polyclonal rabbit anti-SARS-CoV-2 nucleocapsid antibody (1:1,000; Sino Biological, #40143-T62) and a secondary peroxidase-labeled goat-anti-rabbit IgG antibody (1:2000; Dako, #P0448). The signals were developed with a precipitate-forming TMB substrate (TrueBlue; SeraCare/KPL, #5510-0030) and the number of plaques per well was quantified with an ImmunoSpot Image Analyzer (CTL Europe GmbH). The 50% plaque reduction neutralization titer ($PRNT_{50}$) was estimated by calculating the proportionate distance

between two dilutions from which the endpoint titer was calculated. An infection control (without serum) and positive serum control (Nanogam® 100 mg/mL, Sanquin) were included on every assay plate. When no neutralization was observed, the $PRNT_{50}$ was assigned a value of 10.

### Detection of SARS-CoV-2 variant-specific binding antibodies

S-specific binding antibodies were measured by an in-house ELISA as previously described[17] on the same selection of samples that was used to assess the levels of SARS-CoV-2-specific neutralizing antibodies. High-binding EIA/RIA 96-well plates were coated (20 ng/well) with HEK293T cell-generated and His-tagged S1+S2 trimeric S protein (Sino Biological) from ancestral SARS-CoV-2 (D614G; #40589-V08H8), Omicron BA.1 (#40589-V08H26), Omicron BA.5 (#40589-V08H32), or Omicron XBB.1.5 (#40589-V08H45) at 4 °C overnight. Next, the plates washed with 0.05% PBS-T (0.05% Tween-20 [Merck, #P1379] in PBS) and blocked with blocking buffer (PBS-T+2% skim milk powder [wt/vol, Merck, #70166]) at 37 °C for 1 h. The plates were incubated with a 5-fold dilution series of serum starting at a 1:40 dilution (in blocking buffer) at 37 °C for 2 h. After serum incubation, the plates were washed five times with PBS-T, and horseradish peroxidase (HRP)-labeled rabbit anti-human IgG (1:6000; Dako, #P0214) was added. Plates were incubated at 37 °C for 1 h, washed five times with PBS-T, and developed with TMB (3,3′,5,5′-tetramethylbenzidine) peroxidase substrate (SeraCare/KPL, #5120-0047). Absorbance at 450 nm ($OD_{450}$) was measured using an ELISA microtiter plate reader (Anthos 2001 microplate reader) and corrected by subtracting absorbance at 620 nm ($OD_{620}$). A min-max S-curve was subsequently generated based on the lowest (0%) and highest (100%) OD450 value obtained with a reference control consisting of a pool of 19 sera obtained 7 days after bivalent booster vaccination and with high S-specific binding antibody titers (>10,000 BAU/mL). A 50% endpoint titer was calculated by transforming the OD450 values generated per sample by the dilution series to this reference S-curve.

### Detection of T-cell responses by interferon-gamma release assay (IGRA)

The SARS-CoV-2-specific T-cell response was quantified using an interferon-gamma (IFN-γ) release assay (IGRA) in whole blood using the commercially available QuantiFERON SARS-CoV-2 assay kit (QIAGEN, # 626725) as previously described[11]. The assay kit is certified for in vitro diagnostic (IVD) use. Heparinized whole blood was incubated with three different SARS-CoV-2 antigens for 20–24 h using a combination of peptides stimulating both CD4+ and CD8+ T-cells (Ag1, Ag2, Ag3). Mitogen- and carrier (NIL)-coated control tubes were included as positive control and negative control, respectively. After incubation, plasma was obtained by centrifugation, and IFN-γ production in response to antigen stimulation was measured by ELISA (QuantiFERON SARS-CoV-2 ELISA Kit [certified for IVD use]; QIAGEN, # 626420). Results were expressed in international units (IU) IFN-γ/mL after subtraction of the NIL control values as interpolated from a standard calibration curve. LLoD was 0.01 IU/mL and the responder cut-off was 0.15 IU/mL, according to the manufacturer's instructions. Only data obtained with Ag2 (overlapping peptides covering the ancestral S protein) is shown in this manuscript.

### Statistical analysis

A power calculation in the SWITCH-ON trial was performed to identify the number of participants required per study arm, namely: (i) Ad26.COV2.S prime in the DB group, (ii) mRNA-based prime in the DB group, (iii) Ad26.COV2.S prime in the PPB group and (iv) mRNA-based prime in the PPB group. For each arm, 91 participants were required to reach 80% power at a two-sided 5% significance level to detect a difference of 0.2 log10-transformed in the fold change of antibody response between vaccination day and 28 days after boost. This difference was based on the previous HCW study performed at Erasmus MC[39], in which the mean fold changes for adenovirus-primed participants and mRNA-primed participants were reported as 1.344 (SD 0.451) and 1.151 (0.449), respectively.

A descriptive analysis was used to report the baseline characteristics of participants. For continuous variables, mean and standard deviation (SD) were reported if the data have a normal distribution. Otherwise, median and interquartile range (IQR) were used for data with non-normal distribution. Counts and percentages were used to report categorical variables. For missing values, no imputation was performed and data availability was reported in Supplementary Table S4. Immunological data were reported as geometric mean titers or geometric means and 95% confidence intervals. Spearman correlations were reported in Supplementary Fig. S3 and Supplementary Fig. S4. No formal statistical tests were performed to test for differences within or between groups as we deviated from the original protocol in terms of pre-specified outcomes and a lower-than-anticipated sample size[18]. Figures were prepared with GraphPad Prism (v10.2.1) and Adobe Illustrator 2024.

### Reporting summary

Further information on research design is available in the Nature Portfolio Reporting Summary linked to this article.

## Data availability

The data generated in this study are provided in the Source Data file. Materials and samples are available upon reasonable request from the corresponding author and will be released via a material transfer agreement. The SARS-CoV-2 virus stocks are available through the European Virus Archive Global. Accession codes of the viruses used in this manuscript: ancestral (D614G; GISAID: hCov-19/Netherlands/ZH-EMC-2498), Omicron BA.1 (GISAID: hCoV-19/Netherlands/LI-SQD-01032/2022), Omicron BA.5 (EVAg: 010V-04723; hCovN19/Netherlands/ZHNEMCN5892), and Omicron XBB.1.5 (GISAID: hCov-19/Netherlands/NH-EMC-5667). Source data are provided with this paper.

## Code availability

No specific code was written or generated for analysis of the data. Software use has been disclosed.

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

## Acknowledgements

No private funding was received for these studies. The bivalent BA.5 vaccine mRNA-1273.222 was provided by Moderna. Moderna reviewed the final version of the manuscript but had no role in study design, data collection, data analysis, data interpretation, or writing of the report. All other vaccines were supplied by the Center for Infectious Disease Control, National Institute for Public Health and the Environment, the Netherlands (RIVM). This study was funded by the Netherlands Organization for Health Research and Development (ZonMw), grant agreements 10430072110001 (P.H.M.vdK., C.H.GvK., R.D.dV.) and 10430072110008 (C.H.GvK., R.D.dV.). The funder of the study had no role in study design, data collection, data analysis, data interpretation, or writing of the report.

## Author contributions

P.H.M.vdK., C.H.GvK., and R.D.dV. conceptualized the trial. L.M.Z., N.H.T., W.J.R.R., D.G., and R.D.dV. performed the formal analysis. N.H.T., W.J.R.R., R.S.G.S., A.G., D.F.P., L.G.V., V.A.S.H.D., M.L., N.A.K., A.L.W.H., D.vB., M.P.G.K., P.H.M.vdK., C.H.GvK., and R.D.dV. acquired funding. L.M.Z., N.H.T., W.J.R.R., D.G., R.S.G.S., S.B., L.L.A.vD., L.G., L.P.M.vL., S.R., A.G., D.F.P., L.G.V., V.A.S.H.D., M.L., N.A.K., A.L.W.H., B.L.H., D.vB., M.P.G.K., P.H.M.vdK., C.H.GvK., and R.D.dV. were involved in the investigation. L.M.Z., N.H.T., P.H.M.vdK., C.H.GvK., and R.D.dV. performed project administration. A.G., D.F.P., L.G.V., P.H.M.vdK., C.H.G.vK., and R.D.dV. supervised the trial. R.D.dV. visualized the results. L.M.Z., N.H.T., P.H.M.vdK., C.H.GvK., and R.D.dV. wrote the original draft of the manuscript. L.M.Z., N.H.T., W.J.R.R., D.G., R.S.G.S., S.B., L.L.A.vD., L.G., L.P.M.vL., S.R., A.G., D.F.P., L.G.V., V.A.S.H.D., M.L., N.A.K., A.L.W.H., B.L.H., D.vB., M.P.G.K., P.H.M.vdK., C.H.GvK., and R.D.dV. reviewed and edited the final version of the manuscript.

## Competing interests

The authors declare no competing interests.

## Additional information

[1]Department of Viroscience, Erasmus University Medical Center, Rotterdam, the Netherlands. [2]Department of Hospital Pharmacy, Erasmus University Medical Center, Rotterdam, the Netherlands. [3]Center of Tropical Medicine and Travel Medicine, Department of Infectious Diseases, Amsterdam University Medical Centers, Amsterdam, the Netherlands. [4]Infection and Immunity, Amsterdam Public Health, University of Amsterdam, Amsterdam, the Netherlands. [5]Department of Internal Medicine and Infectious Diseases, University Medical Center Groningen, Groningen, the Netherlands. [6]Department of Infectious Diseases, Leiden University Medical Center, Leiden, the Netherlands. [7]Department of Internal Medicine, Division of Allergy and Clinical Immunology, Erasmus University Medical Center, Rotterdam, the Netherlands. [8]Department of Immunology, Erasmus University Medical Center, Rotterdam, the Netherlands. [9]Department of Internal Medicine, Erasmus University Medical Center, Rotterdam, the Netherlands. [10]Department of Experimental Immunology, Amsterdam University Medical Centers, Amsterdam Institute for Immunology and Infectious Diseases, University of Amsterdam, Amsterdam, the Netherlands. [11]Department of Medical Microbiology and Infection Prevention, University Medical Center Groningen, University of Groningen, Groningen, the Netherlands. [12]Center for Infectious Disease Control, National Institute for Public Health and the Environment, Bilthoven, the Netherlands. [14]These authors contributed equally: Luca M. Zaeck, Ngoc H. Tan. [15]These authors jointly supervised this work: P. Hugo M. van der Kuy, Corine H. GeurtsvanKessel, Rory D. de Vries. ✉e-mail: c.geurtsvankessel@erasmusmc.nl

## SWITCH-ON Research Group

Luca M. Zaeck ®[1,14], Ngoc H. Tan[2,14], Wim J. R. Rietdijk ®[2], Daryl Geers[1], Roos S. G. Sablerolles[2], Susanne Bogers[1], Laura L. A. van Dijk ®[1], Lennert Gommers[1] & Leanne P. M. van Leeuwen[1], Sharona Rugebregt[1], Abraham Goorhuis[3,4], Douwe F. Postma ®[5], Leo G. Visser[6], Virgil A. S. H. Dalm[7,8], Melvin Lafeber[9], Neeltje A. Kootstra ®[10], Anke L. W. Huckriede[11], Bart L. Haagmans ®[1], Debbie van Baarle[11,12], Marion P. G. Koopmans ®[1], Anna van de Hoef[2], Isabelle Veerman Roders[2], Nathalie Tjon[2], Karenin van Grafhorst[2], Nella Nieuwkoop[1], Faye de Wilt[1], Sandra Scherbeijn[1], Babs E. Verstrepen[1], Marion Ferren[1], Kim Handrejk[1], Katharina S. Schmitz[1], Koen Wijnans[1], Aldert C. P. Lamoré[13], Jenny Schnyder[3,4], Olga Starozhitskaya[10], Agnes Harskamp[10], Irma Maurer[10], Brigitte Boeser-Nunnink[10], Marga Mangas-Ruiz[10], Renate Akkerman[11], Martin Beukema[11], Jacqueline J. de Vries-Idema[11], Sander Nijhof[11], Frederique Visscher[11], Jopie Zuidema[5], Jessica Vlot[6], Eva Spaargaren[6], Naomi Olthof[6], Annelies van Wengen-Stevenhagen[6], Anouk J. E. de Vreede[6], Jytte Blokland[6], Simone van Mill[6], Vivian W. M. Slagter[6], Kitty Suijk-Benschop[6], Jos Fehrmann-Naumann[6], Daphne Bart[6], Elysia van der Hulst[6], P. Hugo M. van der Kuy ®[2,15], Corine H. GeurtsvanKessel ®[1,15]✉ & Rory D. de Vries ®[1,15]

[13]Department of Information and Technology, Information Management Education & Research, Erasmus University Medical Center, Rotterdam, the Netherlands.

