## [Peer Review File · Nature Communications]

Original COVID-19 priming regimen impacts the immunogenicity of bivalent BA.1 and BA.5 boostersReviewer #1 (Remarks to the Author):

This manuscript, titled "Impact of Original COVID-19 Priming Regimen on the Immunogenicity of Bivalent BA.1 and BA.5 Boosters," explores immune responses, including antibody and T cell responses, across different prime-boost combinations. The study reveals that both bivalent BA.1 and BA.5 boosters generate robust antibody and T cell responses to the ancestor but exhibit limited cross-neutralization against XBB.1.5. Notably, it was shown that different to BA.1 bivalent booster, bivalent BA.5 boosting demonstrates a preference for inducing cross-neutralizing antibody responses to BA.5. Most of the findings in this study are not novel but align with those from other studies. The interesting aspect of this research lies in its well-stratified groups with different priming/boosting regimens widely employed during COVID vaccination, which enables the assessment of distinct immune responses triggered by different strategies. The data then reveals that bivalent booster vaccination with Moderna's vaccines led to a more substantial increase in binding and neutralizing antibodies compared to BNT162b2. After BA.5 boosting, the mRNA-primed groups notably exhibited a broader neutralizing antibody response in contrast to the groups with Ad26.CoV2.5 priming. These results underscore the significance of both the priming strategy and the choice of booster vaccine in fostering an effective immune response. Such insights are important for guiding future COVID-19 vaccination policies, especially in light of emerging viral variants with greater genetic differences from the ancestral viral strain.

However, substantial revisions are necessary to emphasize the key and significant findings in this study.

List of comments

Comment 1: (Figure 1)

- a. In this study, individuals who received at least one booster vaccination were included. This implies that among the participants, some had one booster, some had two boosters, prior to the bivalent booster. It has been reported that the second booster resulted in a stronger and longer-lasting neutralizing antibody response. The different boosting doses may lead to different baseline of immune responses and subsequently responses after bivalent booster in this study. Question arises: is there any variation in the booster vaccination doses among different participant groups?
- b. Biological sex plays important role in shaping immune response, contributing to variations in reactions to viral vaccines. Typically, females develop higher innate, humoral, and cellular immune responses to viral infections and vaccines. In this study cohort, a significant gender bias is evident, with a substantial 77% of participants being female overall. Notably, within the direct boost group, the mRNA-based prime/mRNA1273.214 strategy consisted of 88% female participants, contrasting with the Ad26.CoV2.5 prime/mRNA1273.214 group, which comprised only 65% females. Simultaneously, in the postponed boost group, a higher proportion of female participants was also noted in mRNA priming group compared to Ad26.CoV2.5 priming group. Could this gender bias potentially contribute to the broader neutralizing antibody response to BA.5 observed in the mRNA-based prime group?

Comment 2: (Figure 3)

- a. In this section, the data reveals that mRNA-based priming leads to a higher level of binding antibodies to the ancestor after the bivalent booster compared to Ad26.CoV2.5 priming. However, this elevated antibody level doesn't correspond to a higher level of neutralizing activity against the ancestor, as illustrated in Figure S2. It's crucial to extend the analysis to examine the binding antibody levels to various variants. This will help determine if mRNA-based priming induces a higher level of variant-specific binding antibodies or de novo BA.1/BA.5-specific binding antibodies, potentially resulting in enhanced neutralizing activity against variants.
- b. Figure legends "Antibody and T-cell responses after different original...", however, this figure doesn't include any T cell response data.
- c. While there is no discernible difference in T cell response and neutralizing antibody levels between different priming regimens, it is crucial to integrate the neutralizing antibody and T cell data into the main figure. This becomes particularly significant if the higher levels of binding antibodies against

variants can be detected in individuals who received the mRNA-based priming vaccine.

d. Figure S3, highlighting the distinct immune responses induced by booster vaccines from different manufacturers, can be incorporated into the main figure here. Alternatively, it could be presented as a separate main figure, designated as Figure 4, as it constitutes an important finding in the study. Furthermore, given the variations in baseline immune responses, the fold change, rather than absolute titre, of the binding and neutralizing antibodies before and after bivalent boosters can be a more robust indicator.

Comment 3: (Figure 4)

a. Similar to the previous suggestion for Figure S3, the rate of waning neutralizing antibodies can be assessed by calculating the fold change in antibody titers at Mon3 compared to the titers at D28. Figure S4 is unnecessary, and Figure 4 c, d, e, and f can be presented as supplementary figures. In addition, correlations in Figure S3 were evaluated by Spearman's r , however, the values of r^2 were shown in the figure.

b. Figure 4g illustrates the comparison of variants-specific PRNT50 titers relative to ancestor strain neutralisation between BA.1 booster and BA.5 booster, whereas in figure 4h, it is extended to different priming regime/BA.5 boosting regimes. The legend for both figures is not clear and confusing. Additionally, the labelling on each spiderweb plot is too small and may not be reader-friendly. Enhancing readability can be achieved by relocating the labelling text underneath each plot, consistency with other graphs in this figure. Lastly, the labels for the vaccine from Moderna should be replaced with mRNA1273.214/mRNA1273.222, consistent with other labeling used throughout the manuscript.

Comments 4: (Figure 5)

a. This figure not only described the breadth of neutralising antibodies, but also the T cell response and spike binding antibody levels after breakthrough infection. A modification to the legend is needed to accurately reflect these components.

b. Are these participants infected by BA.5?

c. The immune responses observed in blood may not accurately represent the full response elicited by a viral infection. Studies indicate that during early viral infections, local T cell responses tend to be stronger than those in peripheral blood (<https://doi.org/10.1038/s41590-022-01292-1>, DOI: 10.1164/rccm.201207-1245OC). Comparison of the immunogenicity between mRNA vaccination and virus infection based on immune response in the blood may not be reliable. In fact, the data depicted in this figure strongly indicates that breakthrough infections can elicit strong protective immune response, with comparable spike-specific T cell response and antibody neutralization detected in the blood, similar to that induced by bivalent vaccination. This comparison holds more significance than assessing immunogenicity solely based on binding antibody titers to ancestor spike.

d. It would more readable if a side-by-side comparison between vaccination and breakthrough infection could be integrated into this figure.

Reviewer #2 (Remarks to the Author):

Zaack et al. report in this manuscript the data of the SWITCH-ON trial, which aimed to assess B cell and T cell responses following different vaccination regimens. To this end, the authors studied 434 healthcare workers that received either Ad26.COV2.S or an mRNA-based (mRNA-1273 or BNT162b2) priming vaccination, followed by one or more mRNA-based booster vaccination before being randomized to either the direct boost (DB) group, receiving in October 2022 an Omicron BA.1 bivalent vaccine (BNT162b2 Omicron BA.1 or mRNA-1273.214), or the postponed boost (PPB) group, receiving in December 2022 an Omicron BA.5 bivalent vaccine (BNT162b2 Omicron BA.5 or mRNA-1273.222). They found that booster responses of binding and neutralizing IgG antibodies and T cell, targeting the spike (S) protein of ancestral CoV-2, were comparable after Omicron BA.1 and BA.5 bivalent booster

vaccination and independent of the timing of vaccine administration. Compared to Ad26.COV2.S, mRNA-based priming regimens resulted in slightly higher binding antibody levels, however not in higher neutralizing antibodies or T cell responses. mRNA-based priming followed by BA.5 bivalent booster vaccination resulted in better neutralization of BA.5 and comparable neutralization of BA.1 in comparison to the BA.1 bivalent booster vaccination. Breakthrough infection in the PPB group before bivalent vaccination could be administered resulted in comparatively lower levels of binding antibody titers, although Omicron-neutralizing antibody responses and T cell responses were comparable to post-vaccination responses.

The manuscript is well written, the data are clearly presented, and the methods are straightforward. However, the differences in the different groups studied and the results reported are rather small, which begs question of the clinical relevance of these differences. Also, it is questionable whether these data justify the conclusions of this manuscript, including the one reading "Overall, breakthrough infections before and after vaccination were comparatively poorly immunogenic compared to bivalent booster vaccinations."

Major concerns

1) The authors found that, compared to Ad26.COV2.S, mRNA-based priming resulted in slightly higher binding antibody levels, however not in higher neutralizing antibodies or T cell responses. Was this difference of clinical relevance? The authors should provide data on protection against re-infection (or similar) to address this point.

2) mRNA-based priming followed by BA.5 bivalent booster vaccination resulted in better neutralization of BA.5 and comparable neutralization of BA.1 in comparison to the BA.1 bivalent booster vaccination, but the difference was very discrete. Again, was this difference of clinical relevance? The authors should provide data on protection against re-infection (or similar) to address this point.

3) Breakthrough infection in the PPB group (before the bivalent vaccination could be administered) resulted in comparatively lower levels of binding antibody titers, although Omicron-neutralizing antibody responses and T cell responses were comparable to post-vaccination responses. Again, was this difference of clinical relevance? And does this rather discrete difference justify the conclusion that "Overall, breakthrough infections before and after vaccination were comparatively poorly immunogenic compared to bivalent booster vaccinations.". The authors should provide data on protection against re-infection (or similar) to address this point.

Minor point

a) The authors used an IFN-g release assay (IGRA) to quantify S-specific T cell responses. This method has certain advantages as well as disadvantages, the latter of which might have impacted their findings. The authors should discuss these briefly in their manuscript (e.g. in the shortcomings) and reference other approaches to quantify CoV-2-specific T cells.

Reviewer #3 (Remarks to the Author):

Please note for transparency that I am an employee of Pfizer and am an author on a number of BNT162b2 clinical trial publications.

This is a well written manuscript based on a rapidly prepared study based on the opportunity presented by the staggered availability of the BA.1 and BA.5 based bivalent modRNA COVID vaccines from both Pfizer and Moderna. It would not have been easy to bring this study to fruition during the pandemic period, as noted in lines 109-11, so the authors are to be congratulated for pulling it off,

particularly without industrial support (or maybe that helped?).

This paper contributes useful information to support current WHO and ICMRA advice on updating COVID booster vaccines. I recommend publication. I have a few suggestions just for consideration by the authors.

Line 85. BNT162b2 was jointly developed by Pfizer and BioNTech

Line 98. Might it be appropriate to mention that the Ad26 primary schedule was a single dose, whereas the RNA vaccines required two initial doses. Should this difference be considered as a possible factor in subsequent findings of a difference between participants who received primary Ad26 and primary modRNA vaccines?

Line 107. Although no formal statistical testing was performed, unqualified statements are later made about group differences. Might wording about group differences throughout the paper reflect the uncertainty inherent in the absence of formal testing? Though please note that I don't doubt the greater immunogenicity of the Moderna vaccine.

Lines 100-105 and Figure 1. Figure 1 might suggest that the randomisation took place before primary vaccination, whereas it actually took place just before the boost described in this paper.

I believe that a peer-reviewed version of reference 20 is now available <https://www.mdpi.com/2076-393X/11/11/1711>

Overall a good paper that should be published and my comments are just suggestions for the authors to consider.

Point-by-point response to referees (NCOMMS-23-45606)

Reviewer #1 (Remarks to the Author):

This manuscript, titled "Impact of Original COVID-19 Priming Regimen on the Immunogenicity of Bivalent BA.1 and BA.5 Boosters," explores immune responses, including antibody and T cell responses, across different prime-boost combinations. The study reveals that both bivalent BA.1 and BA.5 boosters generate robust antibody and T cell responses to the ancestor but exhibit limited cross-neutralization against XBB.1.5. Notably, it was shown that different to BA.1 bivalent booster, bivalent BA.5 boosting demonstrates a preference for inducing cross-neutralizing antibody responses to BA.5. Most of the findings in this study are not novel but align with those from other studies. The interesting aspect of this research lies in its well-stratified groups with different priming/boosting regimens widely employed during COVID vaccination, which enables the assessment of distinct immune responses triggered by different strategies. The data then reveals that bivalent booster vaccination with Moderna's vaccines led to a more substantial increase in binding and neutralizing antibodies compared to BNT162b2. After BA.5 boosting, the mRNA-primed groups notably exhibited a broader neutralizing antibody response in contrast to the groups with Ad26.CoV2.5 priming. These results underscore the significance of both the priming strategy and the choice of booster vaccine in fostering an effective immune response. Such insights are important for guiding future COVID-19 vaccination policies, especially in light of emerging viral variants with greater genetic differences from the ancestral viral strain.

Authors' reply: We thank the reviewer for the critical review and acknowledging the importance of both priming and boosting regimens as highlighted by our study.

However, substantial revisions are necessary to emphasize the key and significant findings in this study.

List of comments

Comment 1: (Figure 1)

a. In this study, individuals who received at least one booster vaccination were included. This implies that among the participants, some had one booster, some had two boosters, prior to the bivalent booster. It has been reported that the second booster resulted in a stronger and longer-lasting neutralizing antibody response. The different boosting doses may lead to different baseline of immune responses and subsequently responses after bivalent booster in this study. Question arises: is there any variation in the booster vaccination doses among different participant groups?

Authors' reply: We acknowledge the reviewers' comment, and fully agree that the variable 'exposure history' in study participants makes direct comparison complex. However, as acknowledged by the other reviewers, our cohort is unique in this sense, as we have access to the complete uninterrupted history of all SARS-CoV-2 antigen exposures for all individuals in our study.

To provide optimal transparency, we have now summarized the antigen exposure history for the eight different vaccine-primed groups in the table below. Infections prior to bivalent boost were identified via testing (COVID-19 PCR tests and/or at-home antigen self-tests) as well as the detection of nucleocapsid-specific antibodies.

Supplementary Table S4. Baseline table on the number of antigen exposures per original priming regimen and bivalent booster vaccination.

	Vaccines prior to bivalent boost			Infections prior to bivalent boost		
	2	3	4	0	1	2
mRNA BNT162b2 BA.1	0 (0%)	40 (100%)	0 (0%)	11 (28%)	28 (70%)	1 (3%)
mRNA mRNA-1273.214	0 (0%)	43 (93%)	3 (7%)	14 (30%)	32 (70%)	0 (0%)
vector BNT162b2 BA.1	15 (35%)	28 (65%)	0 (0%)	12 (28%)	29 (67%)	2 (5%)
vector mRNA-1273.214	12 (26%)	35 (74%)	0 (0%)	18 (38%)	29 (61%)	0 (0%)
mRNA BNT162b2 BA.5	0 (0%)	35 (97%)	1 (3%)	10 (28%)	25 (69%)	1 (3%)
mRNA mRNA-1273.222	0 (0%)	31 (94%)	2 (6%)	11 (33%)	22 (67%)	0 (0%)
vector BNT162b2 BA.5	14 (36%)	25 (64%)	0 (0%)	11 (28%)	27 (69%)	1 (3%)
vector mRNA-1273.222	15 (37%)	26 (63%)	0 (0%)	15 (37%)	26 (63%)	0 (0%)

There is indeed a difference in the number of vaccines received by the different groups, which is mainly caused by the fact that the mRNA priming regimes consisted of two priming vaccinations, whereas the Ad26.COVS priming regime was single-dose regimen. However, in all groups, the majority of participants received 3 vaccines, which comprise a ‘prime-prime-boost’ regimen for mRNA-primed individuals, and a ‘prime-boost-boost’ regimen for Ad26.COVS-primed individuals. Consequently, if the effect described by the reviewer were to interfere with our analysis, the Ad26.COVS-primed individuals should actually have higher baseline antibody levels (as they often received two boosts), which is not the case.

Next, to comprehensively address the reviewer comment, we compared all participants (as seen in **Figure 3a,b**) to a selection of participants who received precisely 3 vaccinations and had 1 SARS-CoV-2 infection prior to bivalent boost, which comprises the majority of participants in all groups (**Rebuttal Figure 1, RF1**). The kinetics are largely identical between the exposure-defined selection (**RF1c,d**) and all participants (**RF1a,b**; compare **RF1a** vs. **RF1c** and **RF1b** vs **RF1d**), thus leading to the same conclusion eventually. We have now included the ‘baseline exposure table’ in the manuscript, and emphasized that prior antigen exposure in these cohorts is complex in the discussion of the manuscript.

Lines 223–229: “In this context, it is important to emphasize that, given a number of vaccinations with possibly different vaccines and a varying number of exposures to SARS-CoV-2, the number and type of previous antigen exposures (“exposure history”) in a cohort can be a complex factor to account for at this point in the pandemic. However, as part of the SWITCH-ON study, we are in the

unique position to have access to and account for the complete uninterrupted history of all SARS-CoV-2 antigen exposures for all individuals in our study (Supplementary Table S4).”

Rebuttal Figure 1. Binding antibody responses after bivalent booster vaccination. a-d, Detection of (ancestral) spike (S)-specific binding IgG antibodies after Omicron BA.1 (a,c) or BA.5 (b,d) bivalent booster vaccination at baseline, and 7 days, 28 days, and 3 months post-boost, either including all participants (a,b) or selecting for those that received three vaccinations and had one SARS-CoV-2 infection prior to bivalent boost (c,d). Colors indicate the specific prime-boost regimen (orange = Ad26.COVS.S prime, BNT162b2 Omicron BA.1 or BA.5 boost; yellow = Ad26.COVS.S prime, mRNA-1273.214 or mRNA-1273.222 boost; dark blue = mRNA-based prime, BNT162b2 Omicron BA.1 or BA.5 boost; light blue = mRNA-based prime, mRNA-1273.214 or mRNA-1273.222 boost). Data are shown in box-and-whisker plots, with the horizontal lines indicating the median, the bounds of the boxes indicating the IQR, and the whiskers indicating the range. Bold numbers above the plots represent the respective geometric mean (titer) per timepoint. The line graphs next to each panel depict a time course of the respective geometric mean values with 95% confidence intervals.

b. Biological sex plays important role in shaping immune response, contributing to variations in reactions to viral vaccines. Typically, females develop higher innate, humoral, and cellular immune responses to viral infections and vaccines. In this study cohort, a significant gender bias is evident, with a substantial 77% of participants being female overall. Notably, within the direct boost group, the mRNA-based prime/mRNA1273.214 strategy consisted of 88% female participants, contrasting with the Ad26.CoV2.5 prime/mRNA1273.214 group, which comprised only 65% females. Simultaneously, in the postponed boost group, a higher proportion of female participants was also noted in mRNA priming group compared to Ad26.CoV2.5 priming

group. Could this gender bias potentially contribute to the broader neutralizing antibody response to BA.5 observed in the mRNA-based prime group?

Authors' reply: In order to increase transparency with regards to the gender distribution and to comply with the guidance on Sex and Gender reporting provided by Nature Communications, we have now added an additional paragraph in the Methods section (under “*Study design and participants*”) outlining the female-male ratio among our cohort (which is representative of Dutch healthcare workers according to the Central Bureau for Statistics), and the role of sex or gender in study design. Additionally, we have included a comparison of binding antibodies, neutralizing antibodies against ancestral SARS-CoV-2, and T-cell responses across all timepoints between female and male participants (**Supplementary Figure S6**).

While geometric mean titers for the female participants tended to be higher than for the male participants at the same timepoints for both binding and neutralizing antibodies, we did not observe such a trend for T-cell responses. Considering that this is the case for both participants boosted with an Omicron BA.1 or a BA.5 bivalent vaccine, while we only observe an impact on the breadth of the immune response in individuals who received an Omicron BA.5 bivalent booster dose, we do not believe the biological sex bias to be a significant contributing factor.

Our study was not designed and is not powered to investigate biological sex-based differences in booster vaccination response. Consequently, we would like to refrain from conducting a formal post hoc analysis based on sex, which is also discouraged by Nature communications and the guidance on Sex and Gender reporting.

*Lines 323 – 328: “Sex or gender were not considered in study design. Sex was collected in the study design, and reported in the baseline characteristics table. Our cohort comprises 73.6% female individuals, reflecting the female-dominant gender distribution among healthcare workers in the Netherlands³⁶. A comparison of immunogenicity data from female and male participants is provided in **Supplementary Figure S6**.”*

Supplementary Figure S6. Antibody and T-cell responses after bivalent booster vaccination grouped by sex. a-f, Detection of (ancestral) spike (S)-specific binding IgG antibodies (a,d), ancestral SARS-CoV-2 neutralizing antibodies (b,e), and T-cell responses measured by interferon-gamma (IFN- γ) release assay (IGRA) (c,f) after Omicron BA.1 (a-c) or BA.5 (d-f) bivalent booster vaccination at baseline, and 7 days, 28 days, and 3 months post-boost, grouped by sex (female = triangle; male = circle). Colors indicate the specific prime-boost regimen (orange = Ad26.COVID.S prime, BNT162b2 Omicron BA.1 or BA.5 boost; yellow = Ad26.COVID.S prime, mRNA-1273.214 or mRNA-1273.222 boost; dark red = SARS-CoV-2 infection prime, BNT162b2 Omicron BA.1 or BA.5 boost; light red = SARS-CoV-2 infection prime, mRNA-1273.214 or mRNA-1273.222 boost; dark blue = mRNA-based prime, BNT162b2 Omicron BA.1 or BA.5 boost; light blue = mRNA-based prime, mRNA-1273.214 or mRNA-1273.222 boost). Data are shown in box-and-whisker plots, with the horizontal lines indicating the median, the bounds of the boxes indicating the IQR, and the whiskers indicating the range. Bold numbers above the plots represent the respective geometric mean (titer) per timepoint.

Comment 2: (Figure 3)

a. In this section, the data reveals that mRNA-based priming leads to a higher level of binding antibodies to the ancestor after the bivalent booster compared to Ad26.CoV2.5 priming. However, this elevated antibody level doesn't correspond to a higher level of neutralizing activity against the ancestor, as illustrated in Figure S2. It's crucial to extend the analysis to examine the binding antibody levels to various variants. This will help determine if mRNA-based priming induces a higher level of variant-specific binding antibodies or de novo BA.1/BA.5-specific binding antibodies, potentially resulting in enhanced neutralizing activity against variants.

Authors' reply: We agree with the notion of the reviewer and have consequently performed an extensive analysis on variant-specific binding antibody levels after both Omicron BA.1 and BA.5 bivalent booster vaccination across all timepoints included in our manuscript. To this end, we chose to conduct ELISAs on the same sample

selection as used for the neutralizing antibody tests, using trimeric SARS-CoV-2 spike protein from the ancestral variant as well as Omicron BA.1, BA.5, and XBB.1.5 (**Supplementary Figure S4**). The outcome was incorporated into the results section (lines 181 – 186), and a respective protocol was added to the Methods section of the manuscript (lines 425 – 444).

There are two important observations to be made from the variant-specific antibody data: 1) there is no relative decrease of Omicron XBB.1.5-binding antibodies compared to both ancestral- and Omicron BA.1/BA.5-binding antibodies. This is in contrast to what we (and others) have observed for neutralizing antibodies, suggesting that the epitope changes of the XBB.1.5 S protein disproportionately affect antibody binding of its receptor-binding domain (RBD). 2) we do not observe any preferential increase of variant-specific binding antibodies in mRNA-primed individuals.

Additionally, in collaboration with Menno van Zelm at Monash at Monash University we also investigated the capacity of pre-existing SARS-CoV-2-specific memory B-cells to cross-recognize Omicron sub-variants after Omicron BA.1/BA.5 bivalent booster vaccination, which is currently submitted for publication in a separate manuscript. In those studies, we found that Omicron-only memory B-cells are increased only by a booster vaccination with an Omicron BA.5 bivalent vaccine, but not an Omicron BA.1 bivalent, which fits our observations of the BA.5 bivalent booster inducing variant-specific responses.

Supplementary Figure S4. Breadth of the binding antibody response after bivalent booster vaccination. **a,b**, Detection of binding antibodies targeting ancestral SARS-CoV-2 and Omicron BA.1, BA.5, and XBB.1.5 variants after Omicron BA.1 (**a**) or BA.5 (**b**) bivalent booster vaccination at baseline, and 7 days, 28 days, and 3 months post-boost. Colors indicate the specific prime-boost regimen (orange = Ad26.COVS prime, BNT162b2 Omicron BA.1 or BA.5 boost; yellow = Ad26.COVS prime, mRNA-1273.214 or mRNA-1273.222 boost; dark red = SARS-CoV-2 infection prime, BNT162b2 Omicron BA.1 or BA.5 boost; light red = SARS-CoV-2 infection prime, mRNA-1273.214 or mRNA-1273.222 boost; dark blue = mRNA-based prime, BNT162b2 Omicron BA.1 or BA.5 boost; light blue = mRNA-based prime, mRNA-1273.214 or mRNA-1273.222 boost). **c-d**, Correlations between binding and neutralizing antibody titers against ancestral SARS-CoV-2 (circles) and the Omicron BA.1 (triangles), BA.5 (diamonds) and XBB.1.5 (squares) variants after Omicron BA.1 (**c**) or BA.5 (**d**) bivalent vaccination at 3 months post-boost. Correlations were evaluated by Spearman's r . **e,f**, Radar plots depicting the variant-specific binding antibody titers relative to ancestral SARS-CoV-2 binding (set to 100%) after vaccination with bivalent Omicron BA.1 or BA.5. The plots are grouped either by the administered Omicron BA.1 or BA.5 bivalent booster vaccination (**e**) or the original priming regimen after Omicron BA.5 bivalent vaccination (**f**). Data in panels **a,b** are shown in box-and-whisker plots, with the horizontal lines indicating the median, the bounds of the boxes indicating the IQR, and the whiskers indicating the range. Bold numbers above the plots represent the respective geometric mean (titer) per timepoint. The line graphs next to each panel depict a time course of the respective geometric mean values with 95% confidence intervals.

b. Figure legends “Antibody and T-cell responses after different original...”, however, this figure doesn't include any T cell response data.

Authors' reply: See reply below.

c. While there is no discernible difference in T cell response and neutralizing antibody levels between different priming regimens, it is crucial to integrate the neutralizing antibody and T cell data into the main figure. This becomes particularly significant if the higher levels of binding antibodies against variants can be detected in individuals who received the mRNA-based priming vaccine.

Authors' reply: The reviewer is correct in their assertion that **Figure 3** did not include T-cell data, which we had moved to **Supplementary Figure S2** before submission, alongside the respective antibody neutralization data, and seemingly had omitted to adjust the figure legend accordingly. We agree that it is clearer to report these data alongside the binding antibody data in the main figure and have therefore integrated all panels from **Supplementary Figure S2** back into **Figure 3**.

Additionally, we have now included a more in-depth separation of priming regimens, by identifying individuals who contracted COVID-19 before any vaccination was administered. Independent of subsequent vaccination regimen, individuals that were primed by a SARS-CoV-2 infection as their first antigen exposure had consistently lower binding antibody levels, more comparable to adenovirus-based than mRNA-based priming. We now refer to this in the results (lines 141 – 148) and the discussion. As we employed this as a post hoc analysis, sample numbers can in some cases be low, which is why we believe the complete overview is better suited as a supplementary figure (new **Supplementary Figure S2**).

Lines 141 – 148: “When subdividing the two groups, we excluded participants primed with a SARS-CoV-2 infection before their priming vaccination. These participants with an infection as priming were analyzed separately; we observed kinetics that closely resembled those who received Ad26.COVS priming (Supplementary Figure S2). S-specific binding antibodies of SARS-CoV-2 infection-primed individuals in particular were consistently lower when compared to the mRNA-based priming, while T-cell responses were more comparable. Identification of those who experienced a SARS-CoV-2 infection before the priming vaccination occurred post-hoc and consequently only includes a small number of samples.”

d. Figure S3, highlighting the distinct immune responses induced by booster vaccines from different manufacturers, can be incorporated into the main figure here. Alternatively, it could be presented as a separate main figure, designated as Figure 4, as it constitutes an important finding in the study. Furthermore, given the variations in baseline immune responses, the fold change, rather than absolute titre, of the binding and neutralizing antibodies before and after bivalent boosters can be a more robust indicator.

Authors' reply: We agree with the reviewer. Consequently, we have moved the supplementary figure to the main text (now **Figure 4**) and added fold-changes to improve clarity.

Comment 3: (Figure 4)

a. Similar to the previous suggestion for Figure S3, the rate of waning neutralizing antibodies can be assessed by calculating the fold change in antibody titers at Mon3

compared to the titers at D28. Figure S4 is unnecessary, and Figure 4 c, d, e, and f can be presented as supplementary figures. In addition, correlations in Figure S3 were evaluated by Spearman's r , however, the values of r^2 were shown in the figure.

Authors' reply: We prefer to keep **Figure 4c-f** (now **Figure 5c-f**) in the main manuscript, instead of reporting fold-changes at this point. We believe that this form of visualization allows for an improved assertion by the reader as it does not only put the waning of neutralizing responses into a clear perspective over time, but simultaneously visualizes the non-linearity of the waning of Omicron BA.1/BA.5-neutralizing in comparison to ancestral-neutralizing antibodies. We also think that the addition of correlations of binding and neutralizing antibodies is worthwhile and would prefer to keep it a supplementary figure as it allows independent assertion of data integrity. The Spearman's r is now correctly shown in each panel as indicated by the reviewer.

b. Figure 4g illustrates the comparison of variants-specific PRNT50 titers relative to ancestor strain neutralisation between BA.1 booster and BA.5 booster, whereas in figure 4h, it is extended to different priming regime/BA.5 boosting regimes. The legend for both figures is not clear and confusing. Additionally, the labelling on each spiderweb plot is too small and may not be reader-friendly. Enhancing readability can be achieved by relocating the labelling text underneath each plot, consistency with other graphs in this figure. Lastly, the labels for the vaccine from Moderna should be replaced with mRNA1273.214/mRNA1273.222, consistent with other labeling used throughout the manuscript.

Authors' reply: We understand the reviewer's concerns regarding clarity and readability. Consequently, we have simplified and streamlined **Figure 4g,h** (now **Figure 5g,h**). Instead of showing four subpanels in **Figure 4h** (now **Figure 5h**) detailing the priming-based differences between either Omicron BA.5 bivalent vaccine (mRNA-1273.222 and BNT162b2 Omi BA.5), we opted to focus purely on the priming-based differences themselves. Simultaneously, font sizes were increased and the figure legend adapted to enhance readability and clarity.

Excerpt figure legend for **Figure 4** (now **Figure 5**):

"[...] g,h, Radar plots depicting the variant-specific PRNT50 titers relative to ancestral SARS-CoV-2 neutralization (set to 100%) after vaccination with bivalent Omicron BA.1 or BA.5. The plots are grouped either by the administered Omicron BA.1 (orange) or BA.5 (purple) bivalent booster vaccination (g) or the original priming regimen (vector-based = yellow; mRNA-based = blue) after Omicron BA.5 bivalent vaccination (h). [...]"

Comments 4: (Figure 5)

a. This figure not only described the breadth of neutralising antibodies, but also the T cell response and spike binding antibody levels after breakthrough infection. A modification to the legend is needed to accurately reflect these components.

Authors' reply: The figure legend of **Figure 5** (now **Figure 6**) has been adapted accordingly.

b. Are these participants infected by BA.5?

Authors' reply: While the SARS-CoV-2 infection of the 12 individuals, who contracted COVID-19 before their scheduled vaccination date, was confirmed by at-home antigen testing, the respective variant they were infected with was not determined. Based on the time frame the infection occurred in (between September and December of 2022), the circulating variants in the Netherlands were BA.5 (September to mid-November) and BQ.1 (mid-November to December) according to the National Institute for Public Health and the Environment (RIVM; <https://www.rivm.nl/corona/actueel/virusvarianten>).

To clarify this, the respective paragraph in the Results section was amended by the following statement:

Lines 191 – 193: “While the respective variant of concern the participants were infected with was not determined, the dominant variants at the time were Omicron BA.5 and BQ.1¹⁹.”

c. The immune responses observed in blood may not accurately represent the full response elicited by a viral infection. Studies indicate that during early viral infections, local T cell responses tend to be stronger than those in peripheral blood (<https://doi.org/10.1038/s41590-022-01292-1>, DOI: 10.1164/rccm.201207-1245OC). Comparison of the immunogenicity between mRNA vaccination and virus infection based on immune response in the blood may not be reliable. In fact, the data depicted in this figure strongly indicates that breakthrough infections can elicit strong protective immune response, with comparable spike-specific T cell response and antibody neutralization detected in the blood, similar to that induced by bivalent vaccination. This comparison holds more significance than assessing immunogenicity solely based on binding antibody titers to ancestor spike.

Authors' reply: We agree with the reviewer that the immune response as represented in peripheral blood does not fully recapitulate all aspects of the immune response, particularly tissue-resident responses, which respond rapidly to pathogen encounters and have been highlighted to provide protection from viral reinfection. Regardless, while a definitive correlate of protection against SARS-CoV-2 infection has not been identified, neutralizing antibodies are being regarded as the closest available proxy.

However, we do agree with the reviewer that our current analyses of the immunogenicity of breakthrough infections cannot be interpreted as a ‘good or bad’ boost. Binding antibodies are induced to a less extent by infection compared to vaccination, whereas neutralizing antibodies and T-cells are boosted to a similar extent. We have now revised the results header and section to refrain from drawing conclusions, as proposed by the reviewer.

*Sub-heading within the results section (line 188): “**Breakthrough infections lead to boosting of immune responses**”*

d. It would more readable if a side-by-side comparison between vaccination and breakthrough infection could be integrated into this figure.

Authors' reply: As suggested by the reviewer, we included a side-by-side comparison of the geometric means between vaccination and breakthrough infection within the line graphs for **Figure 5b,c** (now **Figure 6b,c**) to enhance readability. A numerical comparison between the geometric mean titers of binding antibody levels was already included in the results section (lines 196 – 199). We chose not to include this for **Figure 5d** (now **Figure 6d**), as an addition of four more reference lines would in this case rather result in the opposite and decrease readability.

Reviewer #2 (Remarks to the Author):

Zaack et al. report in this manuscript the data of the SWITCH-ON trial, which aimed to assess B cell and T cell responses following different vaccination regimens. To this end, the authors studied 434 healthcare workers that received either Ad26.COV2.S or an mRNA-based (mRNA-1273 or BNT162b2) priming vaccination, followed by one or more mRNA-based booster vaccination before being randomized to either the direct boost (DB) group, receiving in October 2022 an Omicron BA.1 bivalent vaccine (BNT162b2 Omicron BA.1 or mRNA-1273.214), or the postponed boost (PPB) group, receiving in December 2022 an Omicron BA.5 bivalent vaccine (BNT162b2 Omicron BA.5 or mRNA-1273.222). They found that booster responses of binding and neutralizing IgG antibodies and T cell, targeting the spike (S) protein of ancestral CoV-2, were comparable after Omicron BA.1 and BA.5 bivalent booster vaccination and independent of the timing of vaccine administration. Compared to Ad26.COV2.S, mRNA-based priming regimens resulted in slightly higher binding antibody levels, however not in higher neutralizing antibodies or T cell responses. mRNA-based priming followed by BA.5 bivalent booster vaccination resulted in better neutralization of BA.5 and comparable neutralization of BA.1 in comparison to the BA.1 bivalent booster vaccination. Breakthrough infection in the PPB group before bivalent vaccination could be administered resulted in comparatively lower levels of binding antibody titers, although Omicron-neutralizing antibody responses and T cell responses were comparable to post-vaccination responses.

The manuscript is well written, the data are clearly presented, and the methods are straightforward. However, the differences in the different groups studied and the results reported are rather small, which begs question of the clinical relevance of these differences. Also, it is questionable whether these data justify the conclusions of this manuscript, including the one reading "Overall, breakthrough infections before and after vaccination were comparatively poorly immunogenic compared to bivalent booster vaccinations."

Authors' reply: We thank the reviewer for the thorough review of our manuscript and valuable insight regarding vaccine effectiveness. As the critique on the immunogenicity of breakthrough infections was also raised by the other reviewers, we have only factually reported the immunogenicity of breakthrough infections in the revised manuscript, without drawing any conclusions.

Major concerns

1) The authors found that, compared to Ad26.COV2.S, mRNA-based priming resulted in slightly higher binding antibody levels, however not in higher neutralizing antibodies or T cell responses. Was this difference of clinical relevance? The authors should provide data on protection against re-infection (or similar) to address this point.

2) mRNA-based priming followed by BA.5 bivalent booster vaccination resulted in better neutralization of BA.5 and comparable neutralization of BA.1 in comparison to the BA.1 bivalent booster vaccination, but the difference was very discrete. Again, was this difference of clinical relevance? The authors should provide data on protection against re-infection (or similar) to address this point.

3) Breakthrough infection in the PPB group (before the bivalent vaccination could be administered) resulted in comparatively lower levels of binding antibody titers, although Omicron-neutralizing antibody responses and T cell responses were comparable to post-vaccination responses. Again, was this difference of clinical relevance? And does this rather discrete difference justify the conclusion that "Overall, breakthrough infections before and after vaccination were comparatively poorly immunogenic compared to bivalent booster vaccinations.". The authors should provide data on protection against re-infection (or similar) to address this point.

Authors' reply: While we agree with the reviewer that translation of immunogenicity data into clinical relevance is an important key aspect, this would be a "next step" from our study. Our study, SWITCH-ON, is designed and intended as an immunogenicity study to determine the magnitude, durability, and breadth of an additional (bivalent) booster vaccination in a previously vaccinated cohort of healthcare workers. By having such a well-defined and well-described cohort of which we virtually know their entire, uninterrupted sequence of SARS-CoV-2 antigen exposures, we are in a unique position to provide an in-depth view of the immunological effects of bivalent booster vaccinations against the background of different original priming regimen/platforms. As the intended outcome of this study is immunogenicity and not vaccine effectiveness, we do not believe our study is designed, suited, or powered to allow for a post hoc analysis of clinical vaccine efficacy. Consequently, we would prefer to refrain from making any assertions on this matter.

However, we do believe that our conclusions on the immunogenicity of breakthrough infections was over-interpreted, and we have now only factually reported the immunogenicity of breakthrough infections in the revised manuscript, without drawing any conclusions. This point was also raised by the other reviewers. Consequently, we have adapted the wording throughout our manuscript, as indicated earlier.

Minor point

a) The authors used an IFN-g release assay (IGRA) to quantify S-specific T cell responses. This method has certain advantages as well as disadvantages, the latter of which might have impacted their findings. The authors should discuss these briefly in their manuscript (e.g. in the shortcomings) and reference other approaches to quantify CoV-2-specific T cells.

Authors' reply: The reviewer is correct in their assertion that there are multiple approaches to quantify antigen-specific T-cells, each with their inherent individual advantages and drawbacks. In this study, we chose to perform interferon-gamma release assays (IGRAs), as they are a scalable, robust, affordable, and easy-to-perform whole blood assay of SARS-CoV-2-specific T-cell responses without the need for specialized equipment. These were important parameters in our decision process, as IGRAs were performed on all blood samples, collected from participants at all timepoints, in all locations of our multi-center study.

Other commonly used, but more complex T-cell assays are the activation-induced marker (AIM) assay, intracellular cytokine stainings, or ELISpots, which also allow investigation of the phenotype, and even the breadth of T-cell responses by stimulating

cells with variant-specific peptide pools. This is in contrast to the commercially-available IGRA kits, which employ assay tubes pre-coated with an overlapping peptide pool specific for ancestral SARS-CoV-2 S protein.

We have ourselves employed AIM assays previously to investigate variant-specific T-cell responses and to study the phenotype of SARS-CoV-2-specific T-cells. There, we observed no significant difference in variant-specific T-cell responses following vaccination (PMIDs: 37088096, 35113647, 34035118). Furthermore, the performance of IGRAs and ELISpots for SARS-CoV-2 were demonstrated to be comparable in healthy individuals (PMIDs: 34623327; 34907916).

Lines 106 – 110: “As the performance of interferon gamma (IFN- γ) release assays (IGRAs) and ELISpots was demonstrated to be comparable in healthy individuals^{19,20}, we chose to assess S-specific T-cell responses by IGRA to have a scalable, robust, and comparable platform across all university medical centers in our study”

Reviewer #3 (Remarks to the Author):

Please note for transparency that I am an employee of Pfizer and am an author on a number of BNT162b2 clinical trial publications.

This is a well written manuscript based on a rapidly prepared study based on the opportunity presented by the staggered availability of the BA.1 and BA.5 based bivalent modRNA COVID vaccines from both Pfizer and Moderna. It would not have been easy to bring this study to fruition during the pandemic period, as noted in lines 109-11, so the authors are to be congratulated for pulling it off, particularly without industrial support (or maybe that helped?).

This paper contributes useful information to support current WHO and ICMRA advice on updating COVID booster vaccines. I recommend publication. I have a few suggestions just for consideration by the authors.

Authors' reply: We thank the reviewer for the kind words and helpful suggestions to improve the quality of our manuscript.

Line 85. BNT162b2 was jointly developed by Pfizer and BioNTech

Authors' reply: This has been amended accordingly.

Line 98. Might it be appropriate to mention that the Ad26 primary schedule was a single dose, whereas the RNA vaccines required two initial doses. Should this difference be considered as a possible factor in subsequent findings of a difference between participants who received primary Ad26 and primary modRNA vaccines?

Authors' reply: The reviewer is correct in their assertion that the primary Ad26.COVS vaccination regime was a single-dose regimen. However, it is important to highlight that the participants, who were primed with the adenovirus-vectored Ad26COVS, have all received multiple mRNA-based booster doses. As this comment overall refers to a point as raised by reviewer #1 in their first comment, we would like to refer back to our answer there and the additional supplementary table (**Supplementary Table S4**) we have now included to clarify the amount and type of antigen exposures among our cohort.

Line 107. Although no formal statistical testing was performed, unqualified statements are later made about group differences. Might wording about group differences throughout the paper reflect the uncertainty inherent in the absence of formal testing? Though please note that I don't doubt the greater immunogenicity of the Moderna vaccine.

Authors' reply: As indicated for reviewers #1 and #2, we have adapted some of the phrasing in our manuscript. Simultaneously, we have now included fold-changes when comparing the immunogenicity of BNT162b2 Omicron BA.1/BA.5 and mRNA-1273.214/222, as suggested by reviewer #1, which adds an additional measure of comparability, while refraining from formal statistical testing for the reasons indicated in the manuscript.

Lines 100-105 and Figure 1. Figure 1 might suggest that the randomisation took place before primary vaccination, whereas it actually took place just before the boost described in this paper.

Authors' reply: In the sentence before the one mentioned by the reviewer, we highlight that *“HCW received either Ad26.COV2.S or an mRNA-based (mRNA-1273 or BNT162b2) priming vaccination regimen, followed by at least one mRNA-based booster vaccination before inclusion in this study.”*, thus explicitly mentioning that primary vaccination of participants occurred outside of this study. It is also described in the Methods section (*“Randomization and masking”*). We believe this makes it clear that the subsequent randomization to one of the two booster groups took place accordingly.

To avoid ambiguity when assessing **Figure 1**, we have now also included this description in the respective figure legend.

Lines 632 – 637: **“Figure 1. SWITCH-ON trial enrollment.** *A total of 592 healthcare workers (HCW) were screened for eligibility. Before inclusion in this study, HCW received either Ad26.COV2.S or an mRNA-based (mRNA-1273 or BNT162b2) priming vaccination regimen, followed by at least one mRNA-based booster vaccination. Of the 592 HCW, 434 were included and randomized 1:1 to the direct boost (n = 219) and the postponed boost (n = 215) group. Following dropouts, a total of 183 HCW received an Omicron BA.5 bivalent vaccine in the postponed boost group.”*

I believe that a peer-reviewed version of reference 20 is now available <https://www.mdpi.com/2076-393X/11/11/1711>

Authors' reply: The preprint, which we cited in our manuscript, has now been exchanged for the peer-reviewed, published version of it.

Overall a good paper that should be published and my comments are just suggestions for the authors to consider.

Reviewer #1 (Remarks to the Author):

The authors have addressed most of my major concerns and questions. No other comments from me.

Reviewer #2 (Remarks to the Author):

Zaack et al. present a mildly revised of their manuscript reporting the data of the SWITCH-ON trial. As previously mentioned, the manuscript is well written, the data are clearly presented, and the methods are straightforward. However, also as previously stated, the differences in the different groups studied and the results reported are rather small, which begs question of the clinical relevance of these differences.

To my previously voiced major concerns (re-copied here below), the authors responded "As the critique on the immunogenicity of breakthrough infections was also raised by the other reviewers, we have only factually reported the immunogenicity of breakthrough infections in the revised manuscript, without drawing any conclusions." I am unsure whether these formal adaptations to the manuscript made "the differences in the different groups studied and the results reported" more significant than reported in the previous version of the manuscript.

Major concerns (for reference, copied from previous assessment)

1) The authors found that, compared to Ad26.COV2.S, mRNA-based priming resulted in slightly higher binding antibody levels, however not in higher neutralizing antibodies or T cell responses. Was this difference of clinical relevance? The authors should provide data on protection against re-infection (or similar) to address this point.

2) mRNA-based priming followed by BA.5 bivalent booster vaccination resulted in better neutralization of BA.5 and comparable neutralization of BA.1 in comparison to the BA.1 bivalent booster vaccination, but the difference was very discrete. Again, was this difference of clinical relevance? The authors should provide data on protection against re-infection (or similar) to address this point.

3) Breakthrough infection in the PPB group (before the bivalent vaccination could be administered) resulted in comparatively lower levels of binding antibody titers, although Omicron-neutralizing antibody responses and T cell responses were comparable to post-vaccination responses. Again, was this difference of clinical relevance? And does this rather discrete difference justify the conclusion that "Overall, breakthrough infections before and after vaccination were comparatively poorly immunogenic compared to bivalent booster vaccinations.". The authors should provide data on protection against re-infection (or similar) to address this point.

Point-by-point response to referees (NCOMMS-23-45606A)

Reviewer #1 (Remarks to the Author):

The authors have addressed most of my major concerns and questions. No other comments from me.

Authors' reply: We thank the reviewer for their prior critical review and appreciate that we have sufficiently addressed their concerns.

Reviewer #2 (Remarks to the Author):

Zaeck et al. present a mildly revised of their manuscript reporting the data of the SWITCH-ON trial.

As previously mentioned, the manuscript is well written, the data are clearly presented, and the methods are straightforward. However, also as previously stated, the differences in the different groups studied and the results reported are rather small, which begs question of the clinical relevance of these differences.

To my previously voiced major concerns (re-copied here below), the authors responded "As the critique on the immunogenicity of breakthrough infections was also raised by the other reviewers, we have only factually reported the immunogenicity of breakthrough infections in the revised manuscript, without drawing any conclusions." I am unsure whether these formal adaptations to the manuscript made "the differences in the different groups studied and the results reported" more significant than reported in the previous version of the manuscript.

Major concerns (for reference, copied from previous assessment)

1) The authors found that, compared to Ad26.COVS.S, mRNA-based priming resulted in slightly higher binding antibody levels, however not in higher neutralizing antibodies or T cell responses. Was this difference of clinical relevance? The authors should provide data on protection against re-infection (or similar) to address this point.

2) mRNA-based priming followed by BA.5 bivalent booster vaccination resulted in better neutralization of BA.5 and comparable neutralization of BA.1 in comparison to the BA.1 bivalent booster vaccination, but the difference was very discrete. Again, was this difference of clinical relevance? The authors should provide data on protection against re-infection (or similar) to address this point.

3) Breakthrough infection in the PPB group (before the bivalent vaccination could be administered) resulted in comparatively lower levels of binding antibody titers, although Omicron-neutralizing antibody responses and T cell responses were comparable to post-vaccination responses. Again, was this difference of clinical relevance? And does this rather discrete difference justify the conclusion that "Overall, breakthrough infections before and after vaccination were comparatively poorly

immunogenic compared to bivalent booster vaccinations.". The authors should provide data on protection against re-infection (or similar) to address this point.

Authors' reply: As outlined in our initial response, we principally agree with the reviewer that it is important to translate these key immunological findings into clinical applications. However, this is not something that we can address in the SWITCH-ON study, which was designed to determine the magnitude, durability, and breadth of immune responses after an additional (bivalent) booster vaccination in a well-characterized cohort of healthcare workers.

Its intended outcomes did not include vaccine effectiveness and, consequently, is neither suited nor powered to make any assertions on that. Now that we found that the original COVID-19 priming regimen has an impact on the immunogenicity of subsequent booster vaccinations, the consequential next step would be to assess how this affects clinical outcome. To emphasize this further, we have amended the discussion.

Lines 298-302 (discussion): "It is important to emphasize that this study was designed to investigate the magnitude, durability, and breadth of immune responses after an additional (bivalent) booster vaccination in a well-characterized cohort of healthcare workers. Consequently, our study cannot make any assertions on vaccine efficacy or other clinical outcomes of the imprinting-based altered immunogenicity of bivalent booster vaccinations."